# Building-scale flood loss estimation through vulnerability pattern characterization: application to an urban flood in Milano, Italy

Andrea Taramelli[1,2], Margherita Righini[1], Emiliana Valentini[3], Lorenzo Alfieri[4], Ignacio Gatti[1], Simone Gabellani[4]

[1]Istituto Universitario di Studi Superiori di Pavia (IUSS), Pavia, 27100, Italy
[2]Institute for Environmental Protection and Research (ISPRA), Roma, 00144, Italy
[3]Institute of Polar Sciences of the Italian National Research Council (ISP CNR), Roma, 00015, Italy
[4]CIMA Research Foundation, Savona, 17100, Italy

*Correspondence to*: Margherita Righini (margherita.righini@iusspavia.it)

**Abstract.** The vulnerability of flood-prone areas is determined by the susceptibility of the exposed assets to the hazard. It is a crucial component in risk assessment studies, both for climate change adaptation and disaster risk reduction. In this study, we analyse patterns of vulnerability for the residential sector in a frequently hit urban area of Milano, Italy. The conceptual foundation for a quantitative assessment of the structural dimensions of vulnerability is based on the modified Source-Pathway-Receptor-Consequence model. This conceptual model is used to improve the parameterization of the flood risk analysis describing: (i) hazard scenarios definition performed by hydraulic modelling based on past event data (Source estimation) and morphological features and land use evaluation (Pathway estimation); (ii) the exposure and vulnerability assessment which consists of recognizing elements potentially at risk (Receptor estimation) and event losses (Consequence estimation). We characterized flood hazard intensity on the basis of variability in water depth during a recent event and spatial exposure also as a function of building surrounding and buildings intrinsic characteristics as a determinant vulnerability indicator of the elements at risk. In this sense the use of a geographic scale sufficient to depict spatial differences in vulnerability allowed to identify structural vulnerability patterns to inform depth-damage curve and calculate potential losses from meso-scale (land use level) to microscale (building level). Results produces accurate estimates of the flood characteristics, with mean error in flood depths estimation in the range 0.2-0.3 m and provide a basis to obtain site-specific damage curves and damage mapping. Findings show that the nature of flood pathways varies spatially and is influenced by landscape characteristics and alters vulnerability spatial distribution and hazard propagation. At the mesoscale, the 'Continuous urban fabric' Urban Atlas 2018 land-use class with the occurrence of at least 80 % of soil sealing shows higher absolute damage values. At microscale, evidence demonstrated that even events with moderate magnitude in terms of flood depth in a complex urbanized area may cause more damage than one would expect.

# 1 Introduction

Flood risk is not stationary, it hinges on climate variability along with changes in vulnerability patterns of exposed elements (Lal et al., 2012). Climate change and socioeconomic developments strongly affect such natural dynamics, which include changes in the probability or intensity of hazards (Elmer et al., 2010; Cammerer et al., 2013; Cammerer and Thieken, 2013) associated with human-induced environmental changes. Vulnerability of elements at risk and the communities' adaptive capacity variations rely on socioeconomic developments entailing land use and changes in the exposure of people and assets

(Hufschmidt et al., 2005; Bouwer et al., 2010; Meyer et al., 2013; Taramelli et al., 2018). Hence, within flood-prone areas one of the cost-effective ways to manage and adapt to floods are through a flood vulnerability assessment. Vulnerability can be described as 1) multi-dimensional (e.g., physical, social-cultural, socio-economic, and environmental) (Balica et al., 2009); 2) dynamic (i.e., vulnerability changes over time); 3) scale-dependent (i.e., vulnerability can be assessed at various spatial and temporal scales) (Fekete et al., 2010; Lal et al., 2012); 4) site-specific (i.e., the approach is defined by a specific location's

needs) (Vogel, C. and O'Brien, 2004). Thus, it is essential to address the drivers of vulnerability how it increases, and how it is distributed to effectively manage risk. Therefore, understanding, quantifying, and analysing the vulnerability and exposure of physical properties is a prerequisite for designing strategies and adopting an approach for its reduction (Papathoma-Köhle et al., 2019). We focus here on the structural dimension of vulnerability of flood-prone areas that has proven to be key to analysing flood risk. Since the structural vulnerability is defined as the potential of a particular class of buildings or

infrastructure facilities to be affected or damaged under a given flood intensity (Faella and Nigro, 2003), damage by flood hazard depends on the vulnerability of exposed buildings (Schanze J., 2006; Merz et al., 2010). Vulnerability matrices, depth-damage curves and vulnerability indices (Balica et al., 2009; Papathoma-Köhle et al., 2019) or indicators-based methodology (Kappes et al., 2012;) are mainly used methods and approaches for physical vulnerability assessment (Fuchs et al., 2019; Kumar and Bhattacharjya, 2020; Malgwi et al., 2020). The indicator-based methods usually provide a scale of flood

vulnerability to element-at-risk, having the strength of allowing significant factors that contribute to flood vulnerability easily to be understood by potential users. However, though the indicator-based method does not require a significant amount of field flood damage records and empirical studies unlike the quantitative method usually expressed by the depth-damage curve functions, they are not for monetary flood vulnerability assessment (Usman Kaoje et al., 2021). Conversely, depth-damage functions estimate direct and tangible quantifiable damages normally defined by interpolating flooding depth and damage data

of a specific asset, economic sector, or land use category (Nasiri et al., 2016) estimating the potential effects of a given flood depth in the investigated area. There are several damage assessments models which have been based on different approaches and amount of variables. Empirical approaches are developed based on damage data compiled after flood events, while synthetic damage models are expert-based models obtained by a what-if analysis (Thieken et al., 2008; Roberts et al., 2009; Pistrika et al., 2014; Amadio et al., 2019; Arrighi et al., 2020). On one side empirical curves give more accurate actual damage

data, on the other synthetic functions show more transferability and comparability. However, their reliability strongly depends on the quality and quantity of input data used (Molinari and Scorzini, 2017; Englhardt et al., 2019). The problem is further

exacerbated by the lack of information on damage explicatory variables, both hazard and vulnerability related (Molinari et al., 2012; Menoni et al., 2016). Akbas et al., (2009) proposed a specific physical vulnerability curve introducing the concept of probabilistic damage functions and appropriate definition of relevant damage states, assessing temporal and spatial impact

probability uncertainties. As a result of neglected attention to disaster risk impacts in the past, it is not easy or even possible to portray the spatial and temporal patterns of flood damage and losses with reasonable precision. This makes measurement of progress in reducing the disaster risk difficult if not infeasible. Therefore, some authors stressed the need for the use of empirical data from past events to provide powerful analytical tools (Apel et al., 2008; Vamvatsikos et al., 2010; Kreibich et al., 2022). The ideal solution would be a model that is simple, as the empirical ones, but that include quantitative and qualitative

explicative parameters in damage assessment, as the synthetic ones, to capture the full complexity of buildings vulnerability (Schanze J., 2006; Balica et al., 2009; Lal et al., 2012; Morelli et al., 2021). The aim of this paper is to improve structural vulnerability assessment for a flood prone area considering vulnerability to be a composition of the hazard intensity and inherent characteristics of elements-at-risk and their surroundings as sources of information to define the more susceptible and exposed residential buildings to inform depth-damage curve and calculate potential losses and mapping damage. To better

understand potentially damaging natural processes interacting with elements at risk the work is structured according to the modified Source-Pathway-Receptor-Consequence (SPRC) conceptual model (Fleming, 2002). The SPRC model relies on a simple causal chain commonly adopted to understand the flood risk system and the link among processes (Schanze J., 2006). The SPRC model consists of the hazard origin identification as to an event transmitted through a pathway to a receptor with possible negative effects for the receptor depending on their vulnerability and their exposure intensity (Hallegatte et al., 2013;

Taramelli et al., 2015). The SPRC model considers the hazards as the flood water that propagates to the receptor resulting in potential consequences (Hallegatte et al., 2013; Taramelli et al., 2015):

- The 'Source' of a flood is the origin of the hazard, usually an extreme meteorological event (e.g., heavy rainfall) triggering floods;
- The 'Pathway' is the route that a hazard takes to get to the receptors. A pathway must exist for a hazard to be realized;

- The 'Receptor' refers to the entities that may be damaged by the hazard depending on their exposure and susceptibility (e.g., people, property or the environment);
- The 'Consequence' is the harm that results from a single occurrence of the hazard, such as economic, social or environmental that may result from a flood. It may be expressed quantitatively (e.g., monetary value), by category (e.g., high, medium, low) or descriptively (USACE, 2019).

Here vulnerability assesses the susceptibility to harm exposed residential buildings when exposed to the hazard (USACE, 2019; Kreibich et al., 2022) to: 1) address the necessity of an integrated qualitative and quantitative approach to define flood physical (structural) vulnerability; 2) to accurately re-assessed past event flood characteristics; 3) to bridge the gap of vulnerability spatialization using spatially distributed information to obtain vulnerability mapping and related potential damage suitable for giving indications on where and how to reduce risks at local level; 4) to bridge the lack of reliable input data in

particular regarding the properties of the element at risks (i.e., the exposure and susceptibility factors) that also contribute to their vulnerability at building level; 5) to improve the exposure analysis integrating the building surrounding as vulnerability indicator (i.e., morphological features and land use). The article is organized as follows: after introducing the current methods to assess flood vulnerability and applications, Sect. 2 describes the case study. Section 3 addresses in detail the input data used and the analytic four steps of the SPRC approach to accurately assess the structural vulnerability. Section 4 depicts the structural vulnerability patterns characterized and classified in terms of their hazard to fluvial flooding based on data from the 2014 Seveso River flood event (Source and Pathway) and exposure, the developing of site-specific damage curves for the residential sector element at risk at meso (i.e., land use level) and microscale (i.e., building level) and the damage mapping (Receptor and Consequence). Section 5 discusses the results of the study, pointing out the advantages, limitations and future developments and, finally, Section 6 summarizes the derived conclusions.

## 2 The Seveso River and the 2014 flood

The Lombardy region, Italy's economic engine, is particularly vulnerable to flood risk (Carrera et al., 2015). It is indeed the most flood-affected region in terms of financial damage, consequently influencing the national growth and stability. The northern part of the city of Milano (45° 28′, 09° 11′) is frequently flooded by the floodwaters of the Seveso River. 342 floods were reported in the last 140 years (i.e., 2.6 per year) (Becciu et al., 2018). On the 14th of July 2014 after a short and intense storm that dropped more than 60 mm of precipitation within 5 hours, the Seveso River overflowed in few sections along its course and flooded some densely populated areas in the provinces of Como, Monza-Brianza, and Milano. Notably, an area of about 3 km$^2$ in the northern part of the city of Milano was flooded, causing closure of streets and public transports and affecting thousands of inhabitants (http://floodlist.com/europe/seveso-river-floods-milan). The Seveso River overflowed at two sections at Niguarda (via Ca' Granda), flowing out of manholes and generating fountains of water and mud that completely inundated Viale Zara and the whole neighbourhood, frequently hit by similar events. The flood caused serious damage to cars, shops, basements and ground floors of many residential buildings. The Isola neighbourhood, near the historic city centre was also affected, and the area of Piazza Minniti was completely inundated. Throughout the northern part of the city, roads were paralyzed for many hours. Official loss data provided by the Lombardy Region based on Damage Report Form in the frame of the loss compensation by the State, reported a total damage of 27.2 million Euro. Private owners were the most impacted (64 %), followed by infrastructure (18 %), commercial activities (13 %), industrial sector (5 %) and environmental (0.4 %) (Fig. 1).

## 3 Material and Methods

In this section we address in detail the analytic four steps of the SPRC model (Fig. 2a) and the data used to accurately assess the structural vulnerability patterns and to understand the links among flood risk system processes (Fig. 2b):

• Source estimation: we reassessed the "inundation map" replicating the flood characteristics providing the flood water depth extension for the study area affected zones.

    • Pathway estimation: we defined inland attributes that can control and influence the event propagation to define in the following step the exposure.

    • Receptor estimation: we considered the elements at risk to perform the exposure and susceptibility analysis to finally
assess structural vulnerability. Thus, elements that are exposed and susceptible to hazard have been categorized into vulnerability homogenous classes.

    • Consequence estimation: we developed site-specific damage curves for the residential sector at meso (i.e., land use level) and microscale (i.e., building level) considering the 2014 flood event and three diverse flood scenarios assuming return periods of 10, 100 and 500 years (Hasanzadeh Nafari et al., 2013).

In the first three steps we identified significant and suitable vulnerability indicators in urban areas representative of three components that contribute to the vulnerability of the elements at risk, i.e., flood hazard intensity, effect of the surrounding environment and building characteristics to produce a vulnerability map at building scale aggregating the indicators and their weights for the selected physical flood. Whereas, in the last step structural vulnerability patterns is used to inform depth-damage curve and calculate potential losses from meso-scale (land use level) to microscale (building level).

**3.1 Hazard scenarios definition: Source and Pathway estimation**

**3.1.1 Source Estimation**

Polygons of the flooded area for this event were collected through the geoportal of the Regione Lombardia. The main data source is represented by surveys and observations from the affected municipalities. Flood related-hazard map of the 2014 flood event is attained by reconstructing the flood affected area and replicating the flood characteristics (i.e., depth of flood water)
using the Floodwater Depth Estimation Tool version 2 (FwDET) (Cohen et al., 2019) implemented in Google Earth Engine. FwDET identifies the floodwater elevation for each cell within a flooded domain based on its nearest flood-boundary grid cell here derived from the Digital Terrain Model (DTM) of the Lombardy Region with a spatial resolution of 5 by 5 meters (Bocci et al., 2015).

**3.1.2 Pathway Estimation**

Morphological features and land use were also considered important factors in influencing hazard propagation. We investigated endanger urban residential areas located within or near landscape sinks (SI) that are potentially filled in conditions of flooding and inefficient drainage system (Dietrich and Perron, 2006; Dodov and Foufoula-Georgiou, 2006; Nardi et al., 2006; Taramelli and Reichenbach, 2008; Thrysøe et al., 2021). These low-lying areas are defined including the DTM 5x5 m and the buildings' footprint layer into a sequential chain of GIS analysis tools. This model is based on the screening of a DTM for landscape
depressions and their maximum extent when filled up at the capacity before spilling over during a flood while ignoring local

infiltration rates and time, thereby allowing the model to select buildings inside or adjacent to these low-lying areas. As the analysis was focused on residential sector damage, exposure information related only to built-up areas was extracted from Copernicus Urban Atlas 2018. These data are exploited to determine how flood vulnerability could rise as a combination of environmental and climate changes effects (Taramelli et al., 2019). Therefore, the Copernicus High Resolution Layer

Imperviousness Degree 2018 with a resolution of 20 m resampled by nearest-neighbours at 5 m is used to identify most exposed residential buildings. Specifically, the normalized Imperviousness Surface Ratio (NISR) was introduced to obtain a proxy to identify buildings most exposed to hazard amplification due to soil sealing (S.L., as stated in the Land Cover/Land Use product nomenclature of the Copernicus Urban Atlas), Eq. (1).

$$NISR = \frac{\text{Imperviousness Degree}}{\text{Building Footprint Area}} \tag{1}$$

**3.2 Exposure and structural vulnerability assessment: Receptor and Consequence estimation**

At the building level, relevant structural indicators (e.g., the building type, the period of construction, the material type, the maintenance state, and building height) are important for determining the susceptibility due to flooding allowing specific building types classification. The structural and non-structural building attributes were linked to the physical characteristics of the damaging flood events to:

• define building susceptibility and finally assess structural vulnerability applying a geographically-distributed and weight-base procedure at building level, the heuristic approach (Receptor Estimation);

• evaluate the buildings potential damage against flood hazards developing site-specific and damage curves-based at both land use and building levels, the probabilistic approach (Consequence Estimation).

### 3.2.1 Receptors Estimation: Heuristic Approach

By overlapping hazard and exposure maps (i.e., the spatial distribution of the elements at risk), a corresponding hazard class is assigned to each building (see Appendix A). This defines the magnitude of damaging flood assigned to each element at risk. Furthermore, at a residential building level relevant descriptive structural attributes are important for determining the susceptibility due to flooding allowing specific building types classification (Figueiredo and Martina, 2016). The structural type combined with the construction materials determine the strength of the building (Corradi et al., 2015). Age and

maintenance are also indications for the current state of the building. Moreover, an estimation of elements-at-risk costs is fundamental to express potential losses in economic terms. The heuristic approach is based on a simple equal weights assignment procedure (Taramelli et al., 2015). The weights assignment has been given by authors based on an intensive literature review and on data availability and quality. The indicators are identified as significant and suitable vulnerability indicators in urban areas representative of three components, i.e., flood hazard intensity, effect of the surrounding environment

and building characteristics. A total score (Eq. 2) is calculated as the sum of single weights assigned to hazard and pathway

(i.e., for water depth $WD$ on the basis of the level in m of inland flooding raster maps, on sink $SI$ map and on $NISR$ classes) (Table 1) and to each structural and non-structural indicator (i.e., Construction Material Type $MT$, Period of Construction $PC$, Building Status $BS$, Building Height $BH$, Building Type $BT$) composing an element at risk. Weights are assigned from 1 = "no or very low response capacity" to 9 = "high response capacity" against flood (Table 2) (Corradi et al., 2015), and based on literature review (Taramelli et al., 2015).

$$\sum_i = (WD_i + SI_i + NISR_i) + (MT_i + PC_i + BS_i + BT_i + BH_i) \qquad (2)$$

Hence, an Info-Table of residential buildings classified by total score is obtained. Residential buildings were classified in 5 vulnerability classes based on the obtained Info-Table from 'Very Low' to 'Very High' vulnerability (Table 3) using the Jenks Natural Breaks algorithm. With natural breaks classification (Jenks), classes are based on natural groupings inherent in the data maximizing the differences between classes. Hereafter buildings were mapped considering the economic unit value in € m$^{-2}$ based on National Real Estate Observatory (OMI) zone and relative market values quotation (€ m$^{-2}$) table obtained from OMI assigned to each residential building type (BT) on the basis of the building status (BS). Here the building economic unit value has been used for estimating the exposed assets in terms of monetary exposure of the residential buildings in the flooded area assuming solely the structure value excluding the content value. For buildings falling in most vulnerable classes (i.e., class 1 and 2, 'Very High' and 'High') a distinction was made between elements with and without basement (Arrighi et al., 2020; Molinari et al., 2020) assigning a weight of 0 to the building with basement and 1 to the building without basement based on the level of damage estimated by McBean et al., (1988) and Crigg and Helweg (1975). As previously mentioned, the assignment of weights refers to the building's response capacity against flood hazard, meaning the capability or incapability of an object to resist the flood impact. Hence, lower values correspond to lower response capacity (high vulnerable buildings), whereas higher values correspond to higher response capacity (low vulnerable buildings) (Taramelli et al., 2015).

### 3.2.2 Consequence Estimation: Probabilistic Approach

The analysis of the negative effects of different event types on exposed elements is necessary to assess the potential damage of elements at risk. Fundamental to the consequence assessment is the concept of depth-damage functions defined as relations between floodwater depth and corresponding damage. Considering hazard scenarios, the depth-damage functions enable the estimation of expected direct losses, hinged on a spatial representation of flood process patterns and categorized elements at risk (Mazzorana et al., 2014). Although damage functions are generated for a specific building of a given type, they can be assumed as reliable predictors of damage for a group of buildings with similar structural/non-structural characteristics. The probabilistic approach is based on the use of damage model INSYDE (Dottori et al., 2016) implemented in R programming language. The model relies on an analysis of physical damages to buildings considering distinctive land use classes and building characteristic parameters to derive synthetic damage curves for residential buildings. INSYDE model can estimate relative damage (i.e., percentage estimation of losses with respect to the total value of the building) and absolute damage, the

latter considers the unit prices of cost of damage. Here we decide to express the damage in absolute terms to give the monetary measure using the cost per unit of measure (e.g., square meter) applying the model deterministically (i.e., without considering any source of uncertainty). To calculate the damage, we combine the exposure and vulnerability data described above with the 2014 flood depth scenario and considering three more existing diverse flood depth scenarios assuming return periods of 10, 100 and 500 years derived from 1-D and 2-D hydraulic modelling designed by the Municipality of Milano in 2019 for the Governmental Territorial Plan and the Flood Risk Management Plan.

Modelled flood damage on residential buildings falling within the study by the following steps:

- Absolute damage was calculated obtaining the absolute damage function and the site-specific depth–damage curve for the residential sector;

- The annual average loss (AAL) and the exceedance probability (i.e., the probability that a certain damage value will be exceeded within a certain return period) for the residential sector were defined. This value is the expense that would occur in any given year if monetary damages from all hazard probabilities and magnitudes were spread out equally over time, representing the full range of hazard magnitudes and offering a more complete picture of monetary impacts;

- Damage modelling and mapping at mesoscale analysis: new site-specific depth–damage functions are then developed for the residential sector at land use level. Potential maximum damage values for residential sector were attributed to each of the Urban Atlas 2018 land use class (i.e., other roads and associated land, industrial, commercial, public, military and private units, green urban areas, discontinuous medium urban fabric, discontinuous dense urban fabric, continuous urban fabric);

- Damage modelling and mapping at microscale analysis: new site-specific depth–damage functions are then developed for the residential sector at building level. Object-based water levels and damage data were integrated with information on building vulnerability considering most vulnerable buildings falling in class 1 and 2 (i.e., Very High and High vulnerability) as resulted from heuristic approach. Features and building characteristics parameters used as model input data are shown in the Appendix B. Therefore, we built depth–damage functions for two building category subsets based on structural and non-structural characteristics frequency (see Fig. C1 in the Appendix). In each category, a distinction was made between elements with and without basement. Here the functions are expressed by coupling the values of flood depth and damage factor (DF). The DF in the damage curves are intended to span from zero (no damage) to one (maximum damage), through absolute damage values normalization. Finally, absolute damage for each residential building was calculated by dividing total damage by building footprint ($€ m^{-2}$) supplying damage for flooded floors including basement if present, and mapped to have spatially distributed information.

# 4 Results

## 4.1 Hazard scenarios definition: Source and Pathway estimation

### 4.1.1 Source Estimation

Gridded estimates of flood depth were produced for the flood polygons of the event. A first quality check of the output denoted unrealistic values in urban areas, especially in the largest flood area in the municipality of Milano, with large portions of the city affected by flood depths of 2 m or larger (Fig. 3). This is in contrast with data reported by the media, referring to flood depths in the order of 30 cm. This is caused by the use of a flood polygon obtained by linking point observations and manual reports, rather than on a spatially continuous identification, such as those provided by aerial or satellite imagery. In addition,

the large degree of urbanization in such area adds noise to the elevation data and consequently to the estimation of flood depths. Hence, in a following step we recomputed flood depths using a flood polygon where the shape of buildings (Open Street Maps, OSM) was first subtracted. In the resulting product, flood depths are mostly within the foreseen ranges, except for some areas with values above 5m along viale Fulvio Testi, Viale Zara and via Volturno. These are attributed to the construction of the line 5 of the underground train line of Milano, particularly to the stations named Ca' Granda, Istria, Marche, Zara and Isola, which

took place in 2008-2010, at the same time of the survey campaign carried out to produce the DTM through Lidar measurements. After filling the holes of the underground stations in the DTM using neighbouring values, resulting flood depths are within realistic ranges, with mean depth of 19 cm and a 90th percentile of the flooded cells of 42 cm.

### 4.1.2 Pathway estimation

Sinks and at-risk buildings were identified and mapped (Fig. 4a). 1,246 buildings, the 81 % of the inundated residential

buildings during the 2014 flood event, lie within or adjacent sink areas where water can potentially pool during a flood and amplify the hazard especially in case of inefficient drainage system. Based on Urban Atlas land use classes the 16.1% of the inundated built-up area was comprised of continuous urban fabric (where the 81% presents an average degree of S.L. > 80 %), the 82.8 % of discontinuous urban fabric (where the 49 % presents an average degree of S.L. between the 50 % and the 80 %) (Fig. 4b). Considering Copernicus Imperviousness Degree map resampled at 5m (Fig. 4c) and building footprint, the NISR

has been calculated. Relating NISR to Urban Atlas land use classes, most exposed buildings to hazard amplification due to soil sealing fall into continuous (S.L. >80 %) and discontinuous (S.L. 50 %-80 %) dense urban fabric mostly located into areas with estimated water depth ranging between 0 and 0.20 m during the 2014 flood (Fig. 5).

**4.2 Exposure and structural vulnerability assessment: Receptor and Consequence estimation**

**4.2.1 Receptors Estimation: Heuristic Approach**

Exposure analysis consisted in the recognition of potentially residential damaged assets (e.g., information on the location, number and type of elements at risk). 1540 residential building have been thereby classified and mapped according to structural and non-structural features (Fig. 6) including the attribution of economic values to define building susceptibility (Fig. 6f). We obtained an Info-table of residential buildings classified by hazard values derived by the weighting of hazard classes on the base of flood depth thresholds, sink map and NISR values and Total Weights derived by structural and non-structural

features weighting. Total score varies from 9 to 34 showing distinct building response capacity against flooding. Low total score values stands for lower building response capacity or missing data. On the basis of the obtained total score, residential building structural vulnerability is defined and mapped (Fig. 7a) distinguishing 5 classes (i.e., Very Low, Low, Moderate, High, Very High) and including the economic unit value based on their relative market values quotation (€ m$^{-2}$). 81 residential buildings fall in class 1 (0-17); 333 residential buildings fall in class 2 (18-20); 432 residential buildings fall in class 3 (21-

23); 477 residential buildings fall in class 4 (24-27); 217 residential buildings fall in class 5 (28-39) (Fig. 7a). For 415 buildings falling in most vulnerable classes (i.e., class 1 and 2, 'Very High' and 'High' respectively) 185 buildings are characterised by the presence of a basement (Fig. 7d). Results show that most vulnerable buildings are mostly located in the southern highly urbanized part of the affected area, closer to the city centre, i.e., zone OMI C12 and C14. Calculating the average relative market values quotation for each vulnerability class we observed that the residential building falling in class 'Very High'

showed the highest average value (i.e., 4148.2 € m$^{-2}$) (Table 4).

**4.2.2 Consequence Estimation: Probabilistic Approach**

To calculate the damage we combine the exposure and vulnerability data described above with the 2014 flood water depth assuming return periods of 500 , 100 and 10 years (Fig. 8).

Firstly, absolute damage has been calculated obtaining a damage function for the residential sector. Resulting total absolute

damage to residential properties is equal to EUR 105.3 million, 100.8 million and 93 million assuming 500, 100 and 10 years of return period respectively and 62.4 million for 2014 flood. It is noteworthy that here the absolute damage modelled values for the considered return periods are always greater than the 2014 event values (Fig. 9a). As it can be seen by observing the depth–damage functions for the residential sector, the shape of the damage for 2014 flood is steeper until 0.2 m water depth than other functions because mostly vulnerable buildings fell within 0.26-0.50 class. Nearly maximum damage occurs when

water depths exceed 0.5 m. The three flood scenarios' function reach their maximum between 1.5 and 2 m (Fig. 9b). The obtained AAL corresponds to EUR 18.3 million. Whilst the events with the lowest annual exceedance probability are associated with the highest total damages, showing that as probability decreases, damages increase (Fig. 9c). As third step, collected data for each inundated building in Milano were used for developing a site-specific mesoscale depth–damage curve for the residential sector (Fig. 10) and mapping (Fig. 11). At mesoscale absolute damage for each Urban Atlas 2018 land use

class resulted higher for the 'Continuous urban fabric (S.L. : > 80 %)' class for all flood scenarios, and for the 2014 flood likewise (Fig. 10e). The function for the 'Continuous urban fabric (S.L. : > 80 %)' runs up to EUR 54.3 million considering the 500 years return period scenario (Fig. 10c) being steeper in the first meter according to literature (Jongman et al., 2012; Huizinga et al., 2017; Scorzini et al., 2022). As can be seen, the shape of the damage for the 'Continuous urban fabric (S.L. : > 80 %) are similar for the 500 (Fig. 10c) and 100 (Fig. 10b) years return period scenarios. Whereas, for the 10 years return

period scenario (Fig. 10a) and for the 2014 flood (Fig. 10d) the functions are much steeper in the first meter and in the first 0.2 m of water depth respectively being characterized by areas affected by low flood depth (less than 0.2-0.5 m). Therefore, the second highest maximum damages values are found for those residential buildings falling within 'Other roads and associated land' land use class with functions running up to maxima of EUR 27 million considering the 500 years return period scenario, reaching its maximum at about 2.5 m of water depth (Fig. 10c).

A more detailed, microscale analysis was then performed integrating object-based water levels and damage data with information on building vulnerability focusing on most vulnerable buildings falling in class 1 and 2 (i.e., Very High and High vulnerability) as resulted from heuristic approach. Specifically we built site-specific microscale depth–damage curves for two building category subsets based on structural and non-structural indicators frequency distribution (Appendix C) making a distinction between elements with and without basement: (1) BT='detached and semi-detached house'; MT='masonry';

BS='good'; NISR='0.05-0.12; BH=1-5 m; PC=1919-1966; (Fig. 12a); (2) BT='detached and semi-detached house'; MT='concrete'; BS='good'; NISR='0.05-0.12; BH=1-5 m; PC=1919-1966 (Fig. 12b). Considering the first subset the shape of the damage for the buildings with basement are not similar, notably the function for the 10 years return period scenario is much steeper in the first 4 m of water depth. The function for the 2014 flood is much steeper in the first 0.20 m of water depth, albeit its maximum value remains lower compared to the other flood scenarios. Considering the buildings without basement,

the shape of the damage functions for the 10 and 100 years return period scenarios are quite similar, as the functions are much steeper in the first 0.5 m of water depth. However, difference with the 500 years return period becomes significant after the flooding depth exceeds 1 m. The function for the 2014 flood is much steeper in the first 0.30 m of water depth, although its maximum value remains lower compared to the other flood scenarios (Fig. 12a). Figure 12b shows the function for the 2014 flood of the building with basement much steeper in the first 0.30 m of water depth. Nearly maximum damage occurs when

water depths exceed 0.3 m. As one can see, the shape of the damage curve for the 500 year return period differs significantly, as the function is less steep albeit it reaches higher maximum damage values at about 1.5 m. Looking at the buildings without basement, the shape of the damage functions for the 10 and 100 years return period scenarios are quite similar as for the 2014 and the 500 year return period scenario. The latter functions are much steeper in the first 0.20 m of water depth (Fig. 12b). Table 5 compares the flood depths for causing a DF equal to 0.1 for the two residential buildings subsets. In each subset, a

distinction was made between elements with and without basement. Spatial distribution and variability are finally obtained by mapping absolute damage for each residential building calculated by dividing total damage by building footprint ($€ \ m^{-2}$) (Fig. 13).

## 5 Discussion

In this study we described the development and the application of a quantitative method based on the causal chain of the SPRC
focusing on structural vulnerability assessment as a fundamental and dynamic component of flood risk analysis in urban areas.
We consider high vulnerability and exposure the outcome of skewed development process, such as those associate with the
intensity of extreme and non-extreme climate and weather events, morphological features, buildings structural and non-
structural characteristics and land use frequently associates to a rapid urbanization and suburbanization as for large
metropolitan regions. Moreover, vulnerability can be seen as situation-specific and scale-dependent, interacting with a hazard
event to generate risk. Therefore, capacity for risk prevention and reduction may be understood as a series of elements,
measures, and tools directed toward intervention in hazards and vulnerabilities with the objective of reducing existing or
controlling future possible risks (Cardona, 2013) at diverse scale of analysis. We characterized flood hazard intensity on the
basis of variability in water depth during a recent event and spatial exposure also as a function of building surrounding and
buildings intrinsic characteristics as a determinant factor of the element at risk susceptibility and response capacity. In this
sense the use of chosen vulnerability indicators and a geographic scale sufficient to depict spatial differences in vulnerability
from land use to building level allow to identify structural vulnerability hotspots and to inform depth-damage curve for
calculating potential damage. Empirical measures of the 2014 flood event enabled a valid characterization of the hazard system
and the intrinsic, underlying relationships and interdependencies required in the structural vulnerability assessment framework.

### 5.1 Hazard scenarios definition: Source and Pathway estimation

Starting with source estimation, we reassessed the 2014 flood characteristics in a densely urbanized area of Milano obtaining
flood depths values within the foreseen ranges, a worthwhile data for the study of highly flood-prone residential buildings.
This requires knowledge about how the source interactions with the natural and non-natural environment lead to the
amplification or the reduction of hazards. The methodology used produces accurate estimates of the flood characteristics, with
mean error in flood depths estimation in the range 0.2-0.3 m (Cohen et al., 2019). Yet, it enables realistic representation of the
flood extent, with robust performance. Furthermore, it is scalable to larger domains and particularly suitable for coupling with
satellite-derived flood extents. In contrast, applications in urban areas are especially challenging, and may need manual fine-
tuning to overcome a range of issues coming from the flood characteristics, elevation data, building size and underground
structures, among others. Indeed, the environment offers resources for human development at the same time as it represents
exposure to intrinsic and fluctuating hazardous conditions (Lal et al., 2012). Between the 2012 and the 2018 in the Milano
metropolitan region the 58 % of land use changes concerns the urban expansion uptake of agricultural areas (European
Environment Agency, 2018). Thus, we estimated the pathway analysing land use focusing on residential built-up areas
identifying low-lying and impervious surfaces (Fig. 4c). In this zone low-lying areas are already affected by periodic flooding
after heavy rainfall and greater urbanization and sprawl could exacerbate this problem. Thus, buildings adjacent or close to
these areas are considered at greater risk of being flooded (Fig. 4a). As is evident from the results NISR higher values fall

within low-lying area. However, these no longer coincide with higher values of water depth recorded for the 2014 flood event (Fig. 5). Therefore, in areas that are already developed, sink map and NISR class can be used to prioritize areas for better risk mitigation planning purposes.

## 5.2 Exposure and structural vulnerability assessment: Receptor and Consequence estimation

By assessing the exposure we defined the receptors and consequently how vulnerability varies spatially. The heuristic approach
ingests residential building exposure categories, source and pathway estimation outputs, meeting the need to achieve a fast, though approximate, vulnerability estimation at building level (Fig. 7). The majority of the residential buildings fall in classes 'Low' and 'Moderate' (Fig. 7a). Moreover, residential buildings falling in class 'High' and 'Very High' are mostly houses built of masonry between the 1919 and the 1970 in a good state of maintenance, of which most with a basement, and the highest average economic unit value in € m$^{-2}$ (Table 4). These buildings are also located where NISR is higher demonstrating
that is a valid support for quantifying buildings most exposed to hazard amplification especially in dense urbanized areas (Fig. 4c). The method resulted to be valuable as a preliminary vulnerability indicator-based assessment of the potential weaknesses in the structural building system to be used in the consequence estimation. Consequence estimation relates to the probabilistic phase of the structural vulnerability based on different scales of model application to obtain the AAL, which is a rough measure of the absolute "riskiness" of a set of exposures and refers to the long-term expected losses per year (i.e., averaged over many
years), and the site-specific depth–damage curves (such as meso and microscale curves). Overall AAL represents the full range of hazard magnitudes offering a more complete picture of monetary impacts should expect to incur over time. Thereby exposed assets as a function of water level were then translated into absolute damage. To calculate the damage we combine the exposure and vulnerability data with the 2014 flood water depth assuming return periods of 10, 100 and 500 years. Flood depths for different return periods show small differences, due to the relatively flat area where the floodwaters can spread. Localised
spots with flood depths larger than 4 metres occur in depressions being filled by the floodwaters (e.g., metro stations, road underpasses). 'Continuous urban fabric (S.L.:> 80%)' class is mostly affected by low flood depth (less than 0.2-0.5 m) for the 2014 food and 10 years return period than the other two scenarios (Fig. 10). Nevertheless, at mesoscale the 'Continuous urban fabric' Urban Atlas 2018 land-use class with the occurrence of at least 80 % of soil sealing shows higher absolute damage values within the first meter according to existing literature. Beside the DF on built up areas is different for diverse types of
land use (Huizinga et al., 2017; Gabriels et al., 2022) it tends to be steeper within the first meter. Residential buildings falling in the 'Roads and associated land' class likewise show great absolute damage values. Roads accounted for the increase in impervious cover giving a significant impact on natural water systems preventing water infiltration (Figs. 11 and 12). Whereas less values of absolute damage are measured for those building located close to 'Green urban areas' is noteworthy (Figs. 11 and 12). At microscale observing the depth–damage functions for the residential sector the shape of the damage for the 2014
flood is steeper to 0.2 m water depth than other functions. Nearly maximum damage occurs when water depths exceed 0.5 m probably because observing the 'Very High' and 'High' vulnerable building distribution against water depth classes is evident that mostly vulnerable buildings fall within 0.26-0.50 m class. Therefore, we considered two subsets of buildings falling in

most vulnerable classes (i.e., class 1 and 2, 'Very High' and 'High') as resulted from heuristic approach building obtaining site-specific depth-damage curves. Observing Table 5 that compares the flood depths needed for causing a DF equal to 0.1 for the different building types, similar values were found for all buildings 'with basement' for the three flood scenarios, and they increased for buildings 'without basement' structures. A possible explanation for this might be due to a larger residence time of water and to the 'filling effect' occurring in basements. It is important to underline that a DF equal to 0.1 for the 2014 is reached at lower values of flood depths for both the subsets, demonstrating that even events with moderate magnitude in terms of flood depth in a complex urbanized area may cause more damage than it would expect.

In general from the findings some key observations emerge:

- At mesoscale the idea that impervious surfaces amplify flood hazards and cause more relevant direct and tangible damage to structures is reinforced. One consequence of increasing impervious cover in urban areas is that conventional urban stormwater systems with underground piping can be overwhelmed when run-off exceeds the capacity of the system and cause surface flooding as for the city of Milano;

- Structural vulnerability measurement and patterns understanding is improved: we derived where vulnerability occurred and how vulnerability was distributed at local scale in a spatially explicit way, indicating vulnerability hotspots. From the selected list of flood vulnerability indicators, the actual circumstances that determine structural flood vulnerability are site-specific, hazard-dependent, and elements at-risk dependent;

- New insights were given for the development of site-specific local residential depth–damage curves for a more comprehensive description of residential buildings damage processes and parameterization across two spatial scale of analysis. Flood damage modelling on the building level is important to optimize investments for the implementation of flood risk management concepts in urban areas.

## 5.3 Advantages, limitations and future developments

Lombardy, Italy's economic engine, is particularly vulnerable to flood hazard risk, the consequences of which are likely to affect the national growth and stability. A better understanding of vulnerability patterns is important for public budgeting, as well as private resilience choices. The study presents several advantages to support decision makers, as planners and policy makers, in improving their investment strategies for the mitigation and the reduction of flood damages, but also to improve the currently low flood insurance coverage exploring new insurance model trials to sustain insurance system for residential properties (Gizzi et al., 2016). In addition, it helps stakeholders working in emergency planning, to set priorities during a flood event, by the identification of the most vulnerable buildings, due to its intrinsic characteristics. The SPRC model is a valuable tool to support decision makers in better understanding vulnerability drivers, complex processes and their interrelations acting as a guide for interventions, priority measures and resources allocation either in the pre or post- event phases. Examples of pre-flood applications of the outputs may include the development of flood hazard mitigation strategies that outline policies and programs for reducing flood losses, including nature-based solutions or the use of the obtained AAL as an input of a cost-

benefits analysis of prevention and mitigation measures. Thereby, examples of post-event applications of the outputs may include the application of land use planning principles and practices, the allocation of resources for flood-resilient buildings interventions. Therefore, the use of an indicator-based methodology to inform the depth-damage curve allows users to add indicators according to their needs, priorities and data availability. Herein, besides using only one characteristic of the building, valuable vulnerability building indicators based on structural and non-structural attributes and surrounding characteristics that

contribute to the vulnerability of the elements at risk are used to identify vulnerability hotspots. The use of spatially distributed information to obtain vulnerability mapping and related potential damage at land use and building levels gives useful indications on elements that will most likely experience the impact of a flood event and consequently where and how specific intervention of protection and maintenance or particular insurance against flood damage could be required. Moreover, the use of the environmental features characteristics to define building surroundings in the vulnerability assessment play an important

role in capturing the totality of the landscape elements that influence the vulnerability patterns amplifying or diminishing vulnerability of elements at risk. Nevertheless, some limitations can be noted. Considerable amount of data at local scale is needed, however it is not always available to the local authorities or cannot be accessible or easily collected. In Italy the current national and regional databases are often of insufficient quality to support a robust analysis. One of the three main elements to be correlated among hazard, vulnerability, and losses is often missing or too hazy to make an appropriate comparison with

scientific findings (Molinari et al., 2014). Here effects of hazard interactions on the buildings susceptibility and exposure is hazard-intensity specific based on 2014 flood event severity, however more empirical data should be included to provide more powerful analytical tools (e.g., vulnerability curves) (Arrighi et al., 2020). Several studies have shown that estimations based on depth-damage functions may be very uncertain since water depth and building use only represent a fraction of the whole data variance (Merz et al., 2004; Fuchs et al., 2019). Therefore, the lack of loss data or scattered information on the damage

suffered by buildings do not allow a proper comparative assessment between past damage flood events data to those findings obtained by modelled depth-damage functions. As well, better information on past events and damage occurrence and amount would provide more information regarding the building response capacity implying detailed event documentation (Fuchs et al., 2019). Thus, the scoring procedure in this study has been done on the basis of literature having many associated uncertainties. Flood over-prediction can lead to over-engineered schemes for defences (Seenath et al., 2016). A conceivable

future development would be to base the scoring procedure on documentation of other available past events occurred in the study area or of future events collecting and rehabilitating the scattered data on the economic impact of past flood events, including the indemnities paid to insurance policy holders, compensations paid for uninsured residential flood damage and state aid provided to economic entities to foster recovery. As for the latter point, available products and services of the European Copernicus Earth Observation program can be a valid support to bridge this gap. Specifically, Copernicus

Emergency Management Service that could provide information about the damage grade, its spatial distribution and extent (i.e., grading map) derived from images acquired in the aftermath of the flood event. Furthermore, the approach could certainly be broadened to commercial buildings and critical facilities (e.g., schools, hospitals, public buildings etc.) enlarging the dataset and the study area.

**6 Conclusion**

The goal of this study was to analyse structural vulnerability patterns to enhance the parameterisation of the flood damage assessment model for the residential sector in a flood-prone area. Data associated to past damaging floods have been elaborated for the purpose of evaluating the vulnerability of a portion of the urbanized area of Milano in relation to the flood intensity. At building level structural vulnerability classes stood strictly driven by the structural type combined with the construction materials, building age and basement presence. On the basis of the 5 structural vulnerability classes the economic unit value

based on their relative market values quotation describes that most exposed buildings, mostly located in the southern highly urbanized part of the affected area, are the one with the highest average market value. Besides buildings attributes, findings indicate that also extrinsic parameters describing the building's surroundings, such as morphological features and land use, indirectly affected vulnerability spatial distribution highly influencing hazard propagation. Results provide a basis to obtain residential building-scale flood damage estimation and site-specific damage curves and mapping. Decision makers, urban and

emergency planners and users that deal with flood insurance might be potential end users of these curves. Good strategies and actions against disasters should be a result of better understanding of disaster risk, thereby understanding where the most vulnerable areas are located considering the level of damage in different parts of the city could have several potential uses such as:

- to produce a more accurate early warning system (EWS);
- to accurately define evacuation sites;
- to prioritize areas for better risk mitigation planning;
- to improve the currently low flood insurance coverage;

This aspect of the research suggested that our results provide evidence for an integrated flood risk management that should consider the entire flood risk system and the interaction among each process. Once SPRC is determined, such a model can be

adopted to:

- Characterizing vulnerability patterns of urban flood damaging event within the bounds of this study at suitable spatial resolution;
- Evaluating impacts at building and landscape scale as a consequence of human interventions on river basins (e.g., river training, loss of flood plains and the retention capacity, the increase of impervious surfaces, large changes of
land cover and intensified land use in particular for the development of settlements);
- Significantly supporting decision-making processes based on cost-efficiency to prioritize effective interventions for flood risk reduction and mitigation boosting the change from the paradigm of flood protection to the paradigm of flood risk management;
- Minimizing the flood risk through changing any of the four elements of the conceptual model.

**Appendix A**

Hazard classes for flood depth on the basis of the level in m of inland flooding raster maps were assigned to residential buildings using the ZONAL STAT tool considering the maximum value (i.e., the largest value of all cells in the value raster that belong to the same zone as the output cell) on the basis of flood depth and velocity thresholds used for the hazard classification in the definition of Lombardy Region Territorial Coordination Plan (Fig. A1).

**Appendix B**

We assumed default values for missing variables (Table B1 and B2).

**Appendix C**

Structural and non-structural building characteristics frequency distribution has been calculated according to vulnerability classes obtained from the heuristic approach (Fig. C1).

**Author contribution**

AT supervised and acquired the financial support for the project leading to this publication. EV and MR developed the methodology. MR, LA, SG and IG analysed data and performed the models. MR and IG prepared the manuscript draft with contributions from all co-authors. AT, EV, MR, LA, IG reviewed and edited the manuscript.

**Acknowledgements**

The authors would like to thank the three anonymous reviewers for their careful reading and valuable comments and suggestion.

**Financial Support**

This research has been supported by EFLIP, a project funded by Fondazione CARIPLO (grant nos.2017-0735).

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

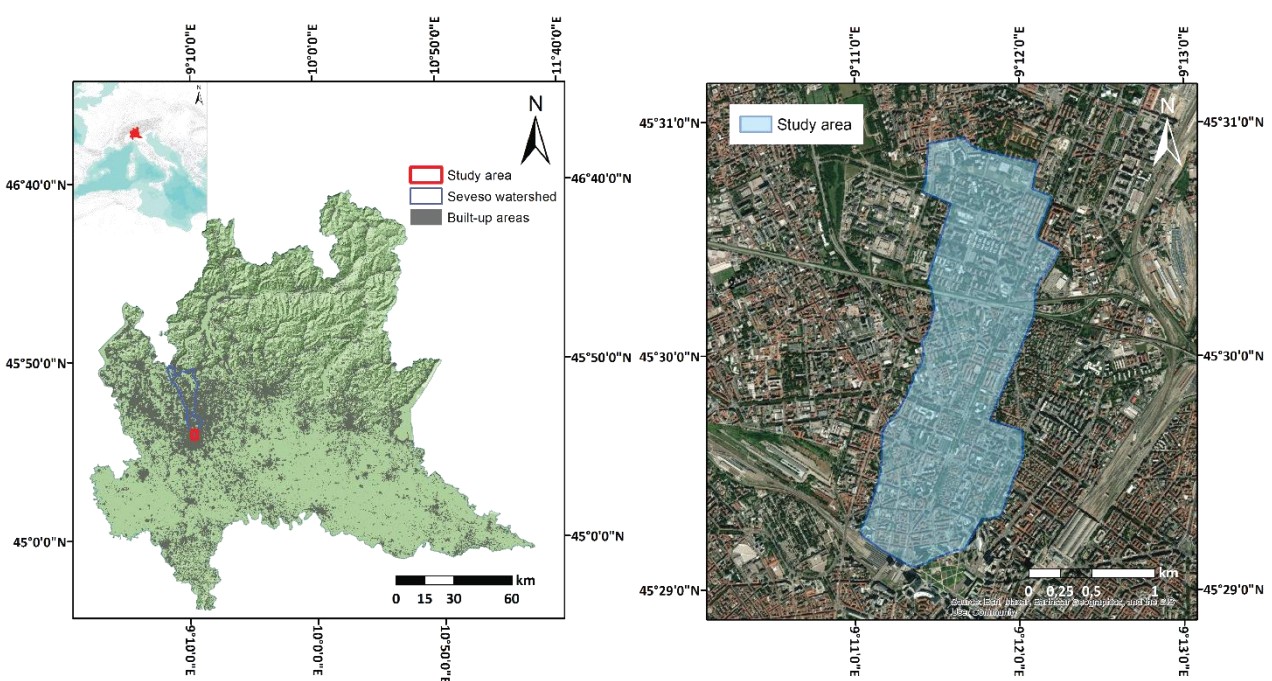

**Figure 1: Investigation area and survey of inundated area of 2014 Seveso flood in Milano. Base map and DTM from © Regione Lombardia 2022. Built-up areas shapefile is from © OpenStreetMap contributors 2022. Distributed under the Open Data Commons**
**Open Database License (ODbL) v1.0. Satellite image is from © Google Earth 2022.**

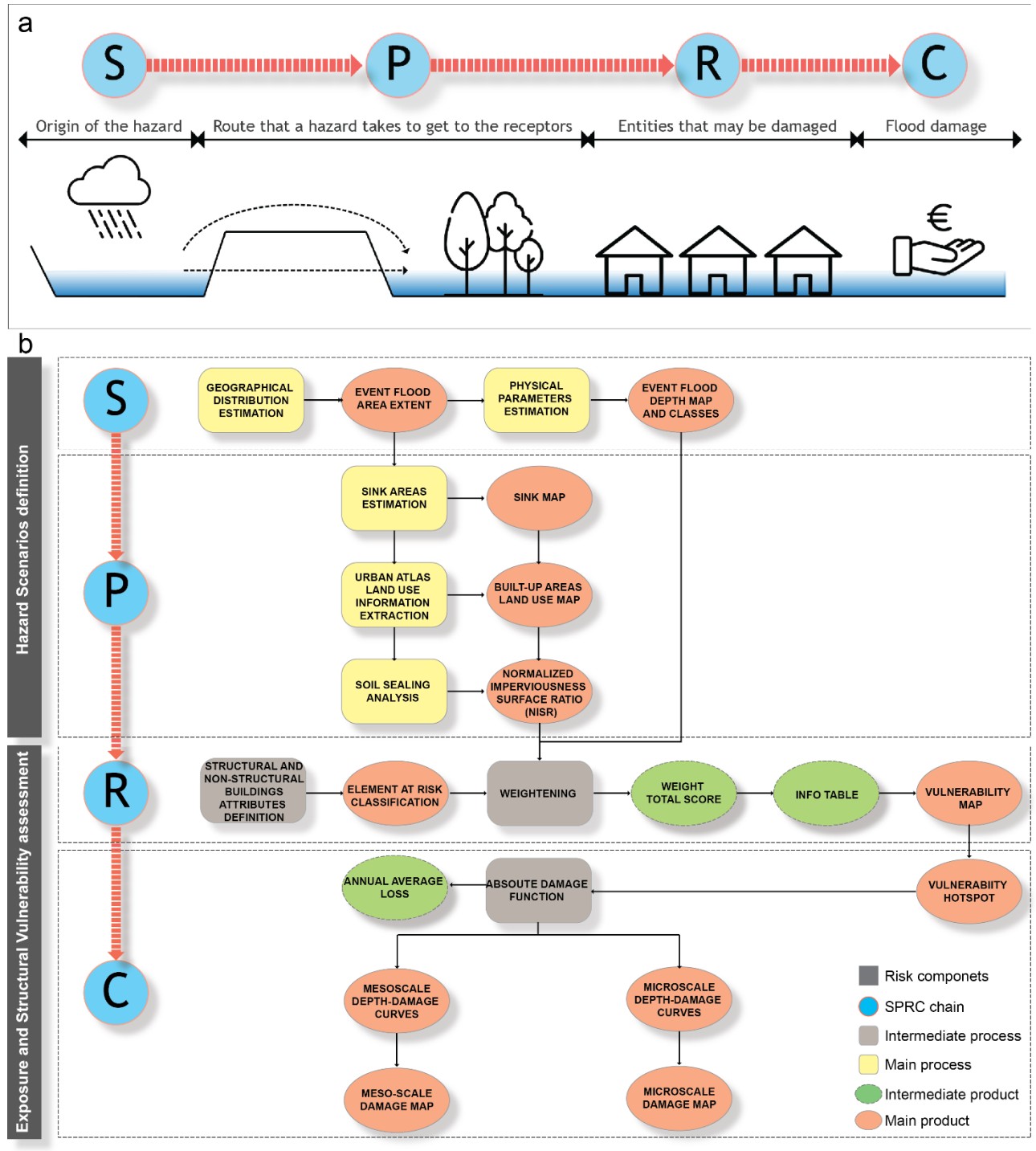

**Figure 2: (a) SPRC conceptual model;(b)Structural vulnerability assessment procedure overview using a modified SPRC model.**

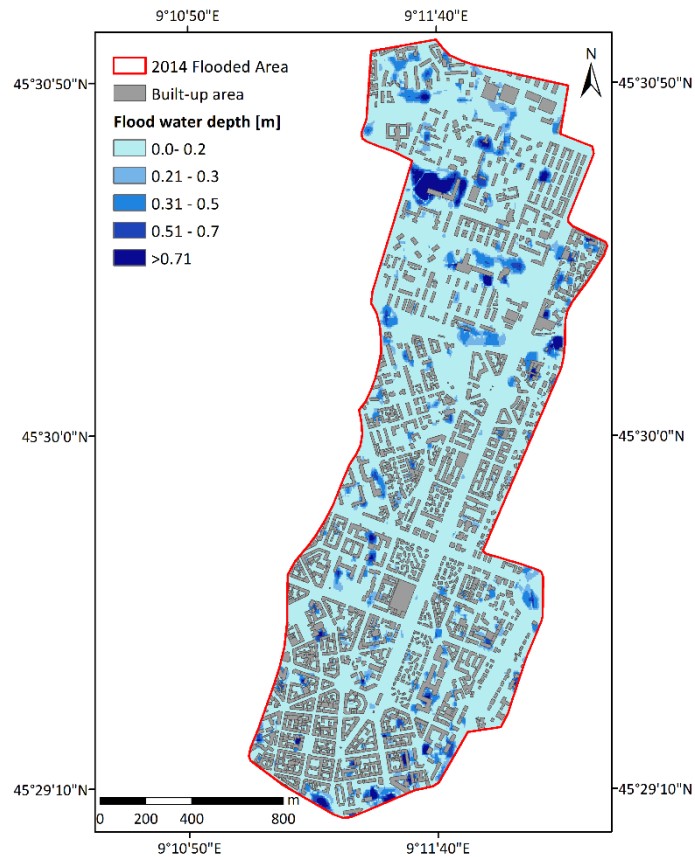

**Figure 3: Estimated flood depths for the flooded polygon in Milano. Built-up areas shapefile from © OpenStreetMap contributors 2022. Distributed under the Open Data Commons Open Database License (ODbL) v1.0. Base map is from © Regione Lombardia 2022.**

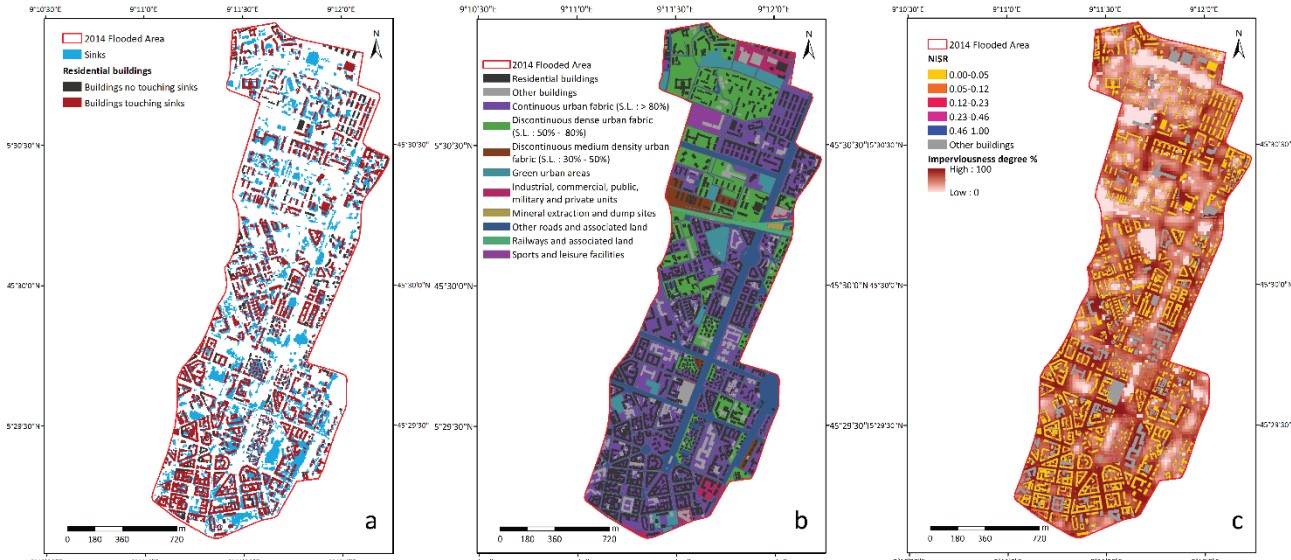

**Figure 4: (a) Sinks distribution map (touching sinks in red; non touching sinks in grey); (b) Urban Atlas 2018 map (residential buildings in black; other buildings in grey (c) Copernicus Imperviousness Degree map resampled at 5m and NISR distribution for residential building. Built-up areas shapefile is from © OpenStreetMap contributors 2022. Distributed under the Open Data Commons Open Database License (ODbL) v1.0. Base maps in (b) and (c) are from © European Union, Copernicus Land Monitoring Service 2022, European Environment Agency (EEA).**

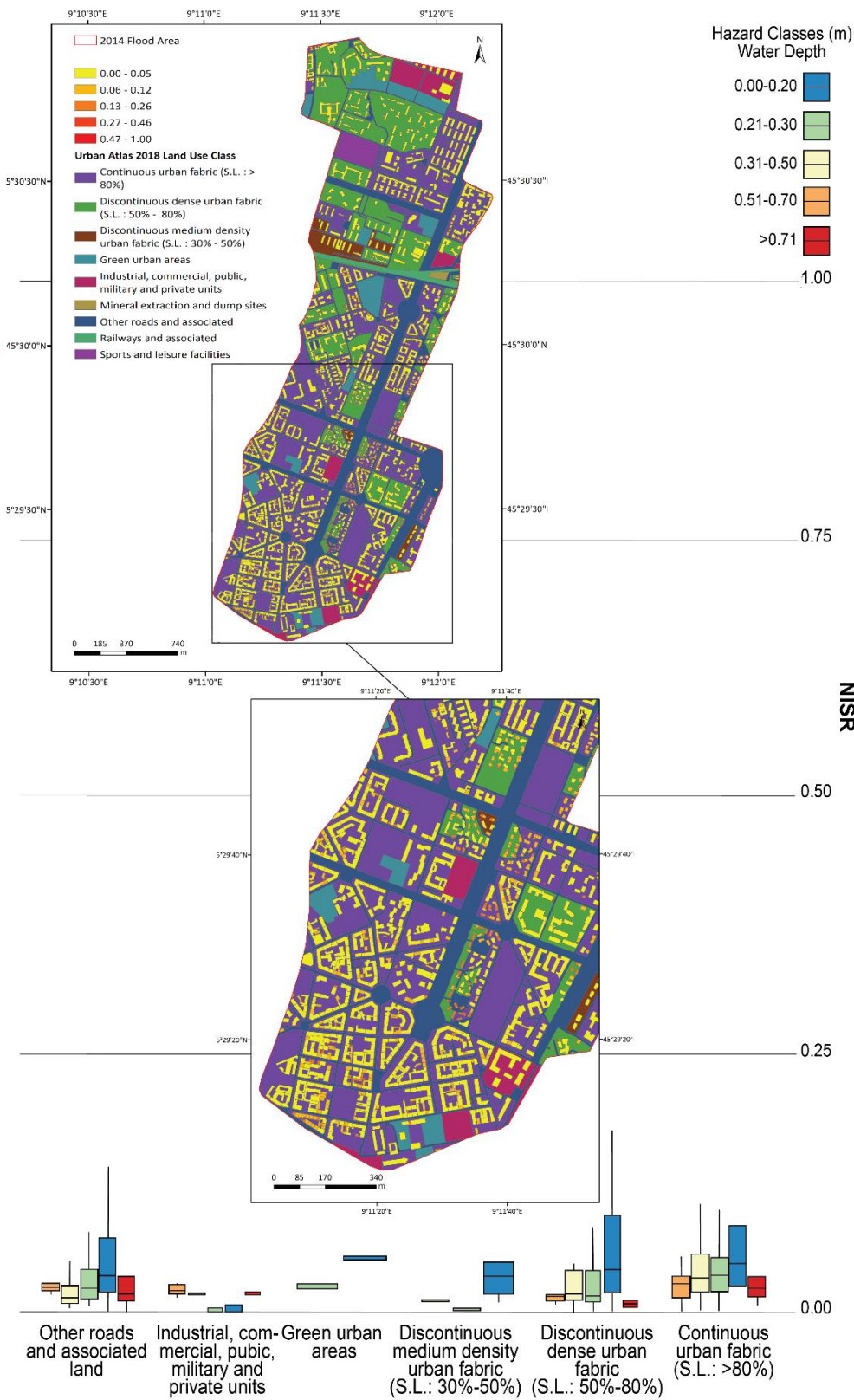

**Urban Atas 2018 Land Use class**

 **Figure 5: Boxplots and maps of NISR distribution for residential building according to Urban Atlas 2018 land use classes and water depth hazard classes. Built-up areas shapefile is from © OpenStreetMap contributors 2022. Distributed under the Open Data Commons Open Database License (ODbL) v1.0. Base map is from © European Union, Copernicus Land Monitoring Service 2022, European Environment Agency (EEA).**

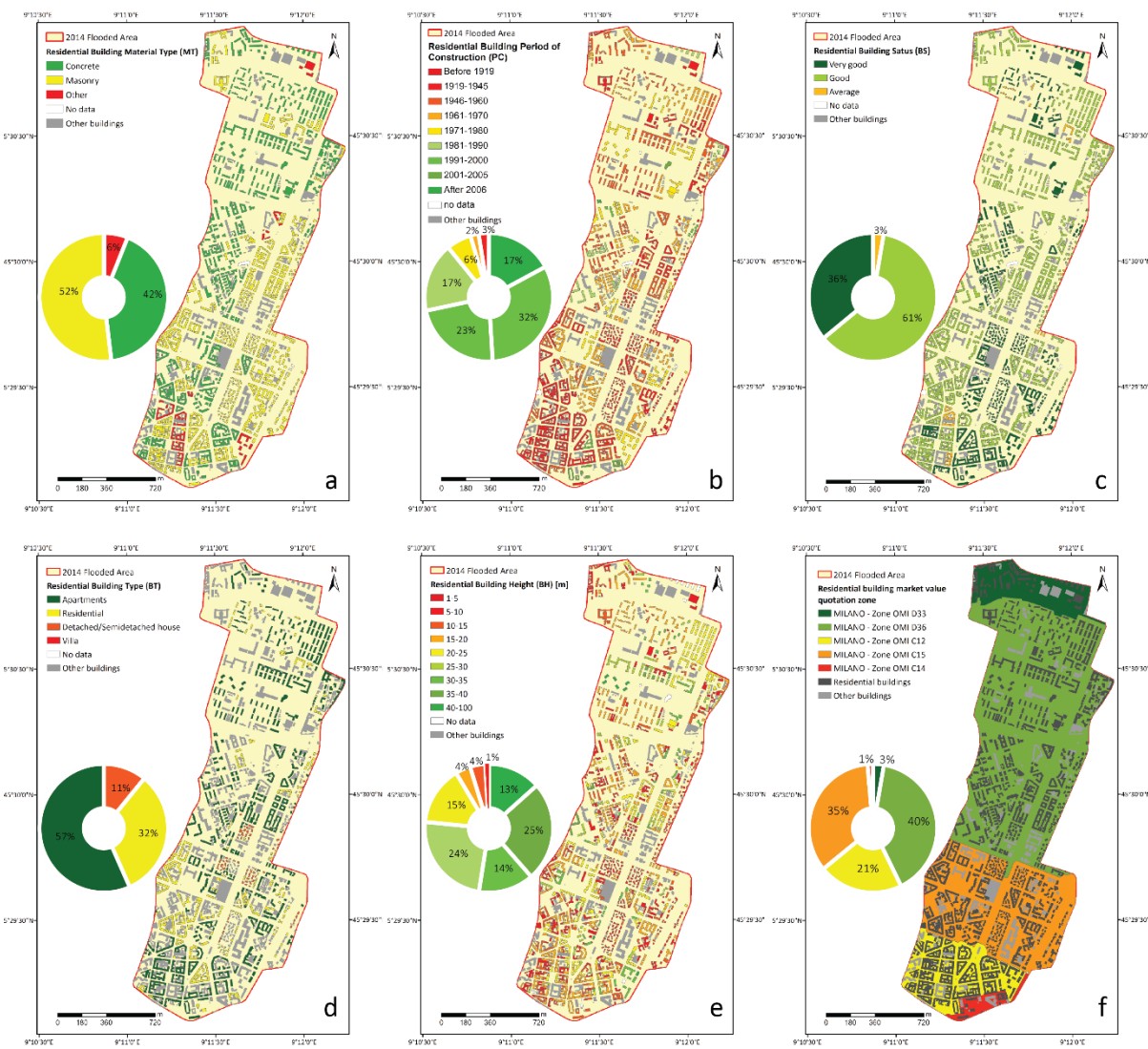

 **Figure 6: Residential Buildings: (a) material type map (MT); (b) period of construction map (PC); (c) building status (BS) map; (d) building type map (BT); (e) building height map (BH); (f) OMI zones map according to the National Real Estate Observatory (OMI) classification based on territory sub-division (central and semi-central urban areas, suburban, peri-urban and peripheral areas) having higher market values quotation (increasing from red to green). Pie charts represent the percentages of residential buildings distribution according to structural and non-structural feature. Built-up areas shapefile is from © OpenStreetMap contributors 2022. Distributed under the Open Data Commons Open Database License (ODbL) v1.0. Base map in (f) is from © Geopoi, Map Data 2022.**

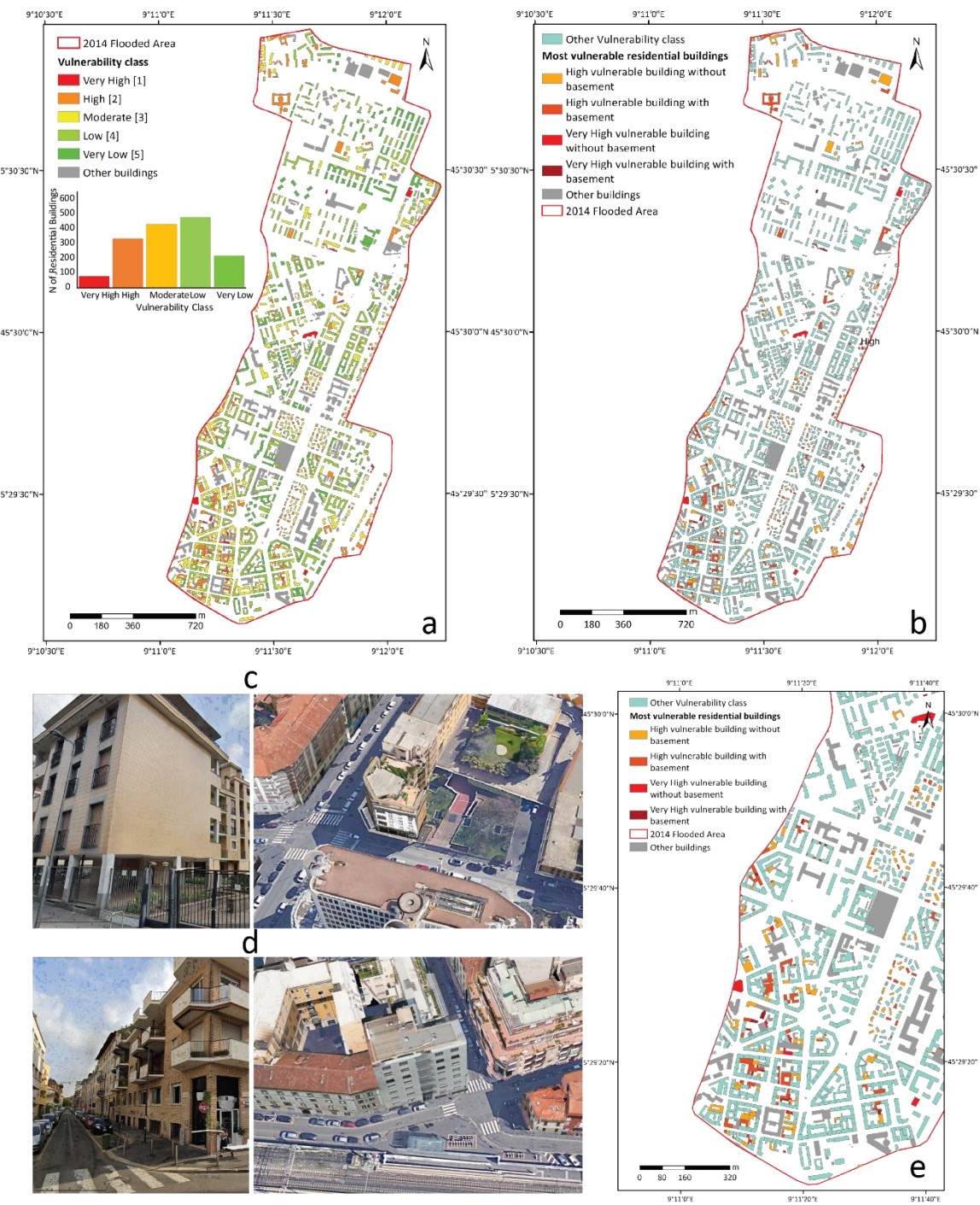

**Figure 7: (a) Residential buildings maps classified by vulnerability class and frequency; (b) most vulnerable residential buildings with and without basement map; examples of residential buildings falling in class High and Very High (c) without basement and (d) with basement respectively and (e) their location. Built-up areas shapefile is from © OpenStreetMap contributors 2022. Distributed under the Open Data Commons Open Database License (ODbL) v1.0. Building pictures are from © Google Maps 2022.**

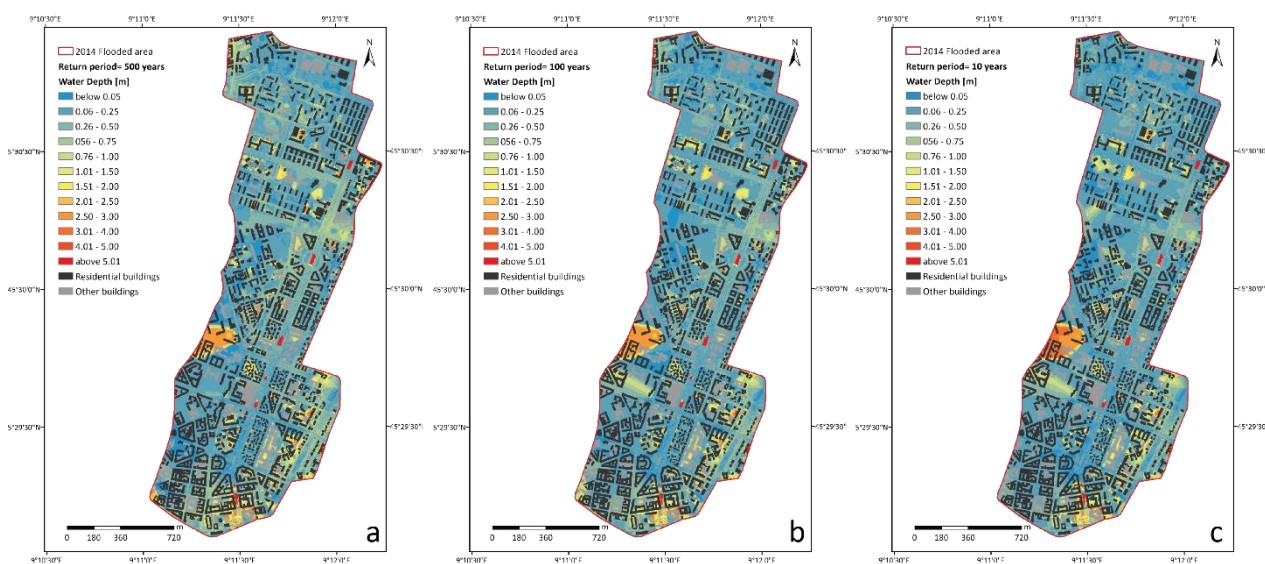

**Figure 8: Estimated water depth and probability of flooding assuming (a) 500, (b) 100 and (c) 10 years of return period scenarios derived from 1-D and 2-D hydraulic modelling designed by the Municipality of Milano in 2019 for the Governmental Territorial Plan and the Flood Risk Management from © Regione Lombardia 2022. Built-up areas shapefile is from © OpenStreetMap contributors 2022. Distributed under the Open Data Commons Open Database License (ODbL) v1.0.**

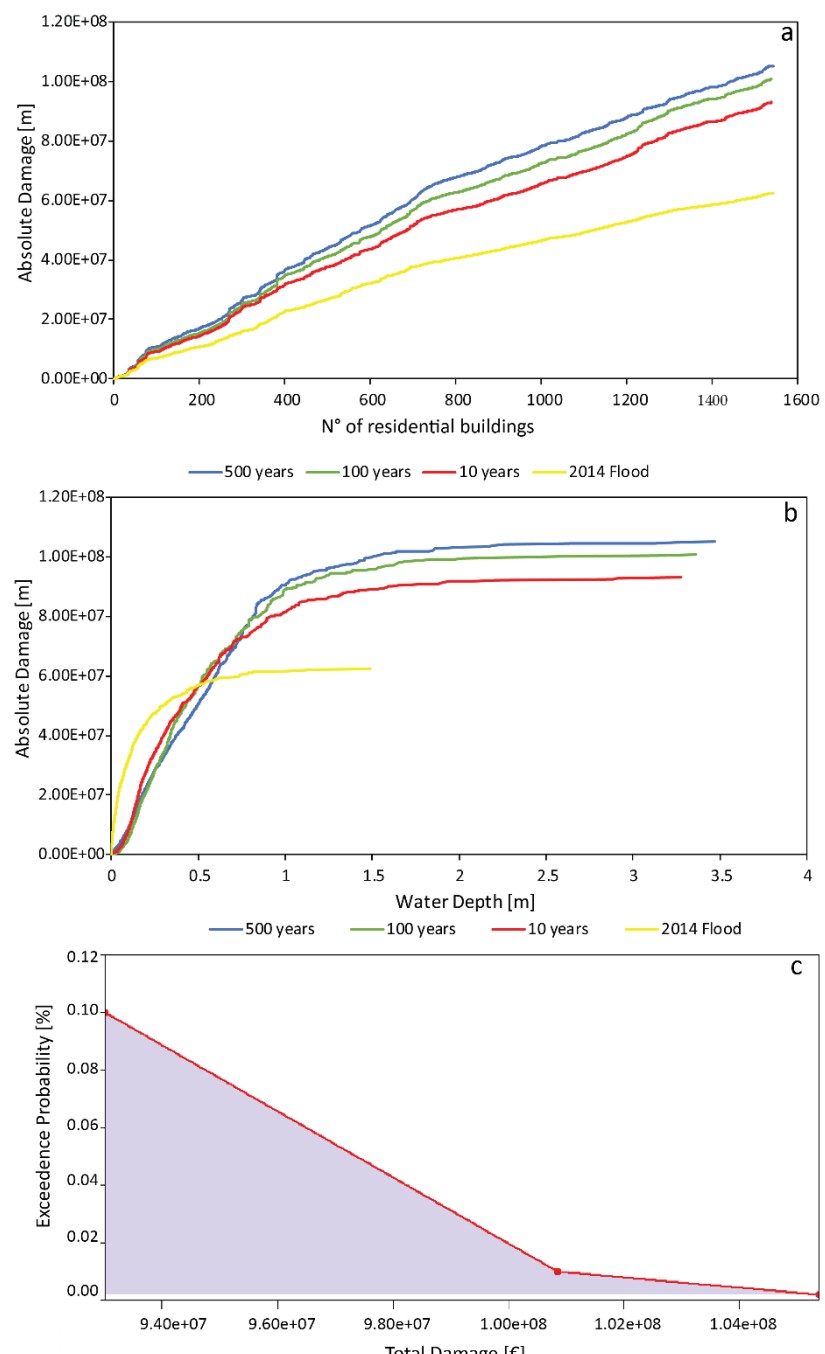

**Figure 9: (a) Absolute Damage for the residential sector (n° of buildings) considering 500, 100 and 10 years of return period and the 2014 flood event; (b) Site-specific depth–damage curve for the residential sector considering 500, 100 and 10 years of return period and the 2014 flood event; (c) Exceedance probability of Absolute Damage for the residential sector.**

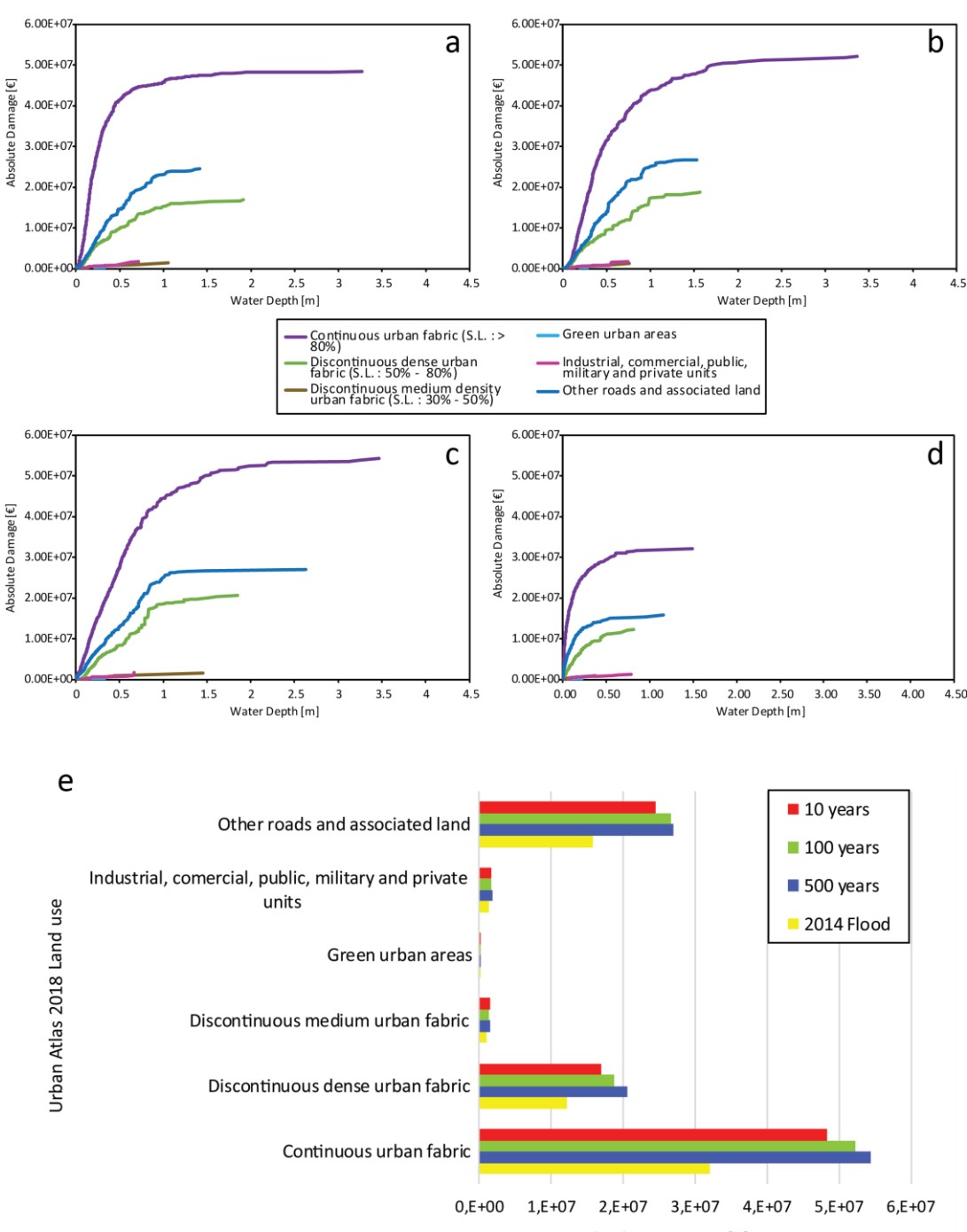

**Figure 10: Site-specific mesoscale depth–damage curves for the residential sector. The x-axis represents the inundation depth; the y-axis represents the damage fraction corresponding to the inundation depth for a specific land use class assuming (a) 10, (b) 100, (c) 500 years of return period and (d) 2014 flood. (e) Comparison of total absolute damage distribution at mesoscale.**


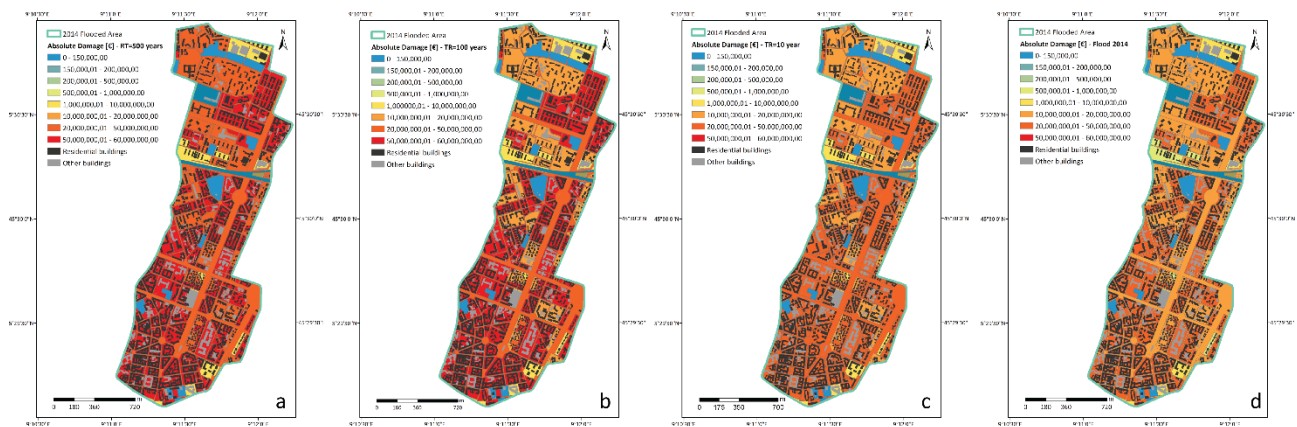

**Figure 11: Mesoscale absolute damage mapping considering (a) 500, (b) 100 and (c) 10 years of return and (d) the 2014 flood event for 2018 Urban Atlas land use classes. Built-up areas shapefile is from © OpenStreetMap contributors 2022. Distributed under the Open Data Commons Open Database License (ODbL) v1.0. Base maps are from © European Union, Copernicus Land Monitoring Service 2022, European Environment Agency (EEA).**


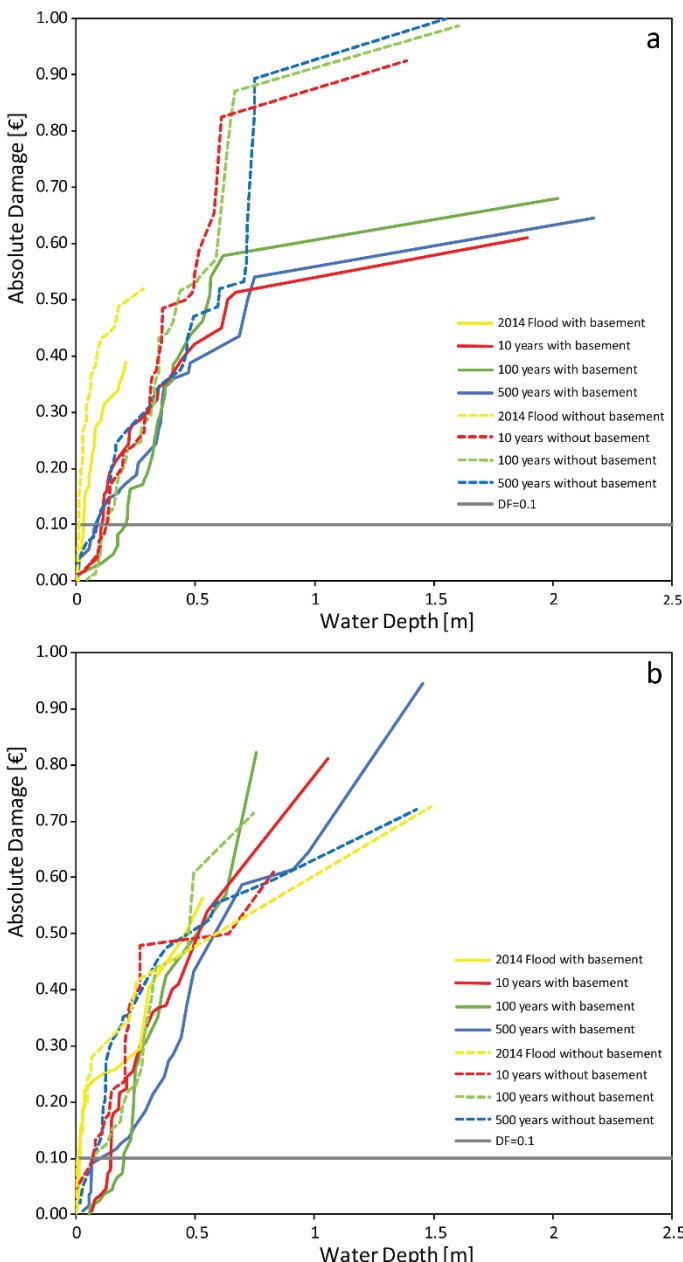

**Figure 12: Site-specific microscale depth–damage curves for the residential sector for the 2014 flood (yellow lines), 10 (red lines), 100 (green lines) and 500 (blue lines) years of return period scenarios for buildings falling in most vulnerable classes (i.e., class 1 and 2, 'Very High' and 'High'): (a) BT='detached and semi-detached house'; MT='masonry'; BS='good'; NISR='0.05-0.12; BH=1-5 m; PC=1919-1966 with basement (solid line) and without basement (dashed lines); (b) BT='detached and semi-detached house'; MT='concrete'; BS='good'; NISR='0.05-0.12; BH=1-5 m; PC=1919-1966 with basement (solid line) and without basement (dashed lines). Solid horizontal grey line stands for DF equal to 0.1.**


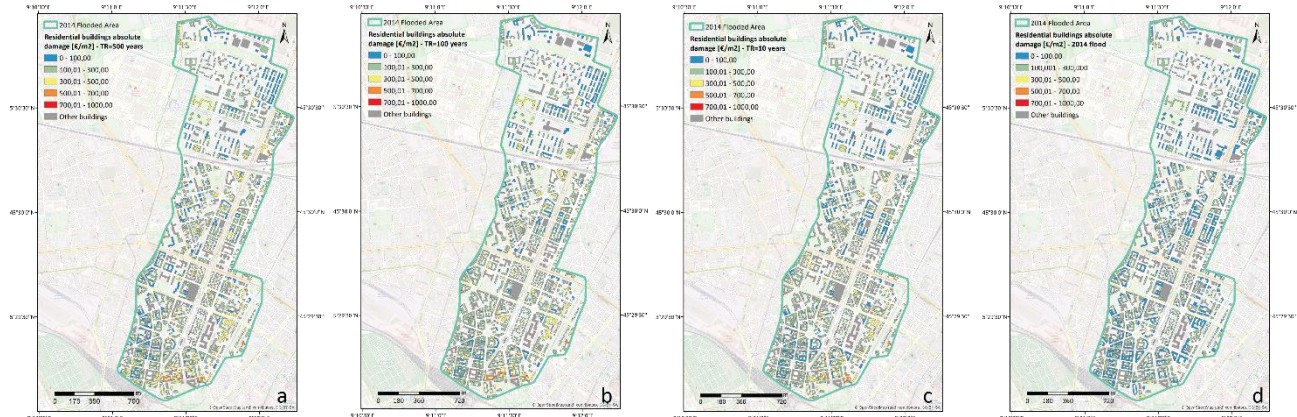

**Figure 13: Microscale absolute damage mapping considering (a) 500, (b) 100 and (c) 10 years of return period scenarios and (d) the 2014 flood event for the two most vulnerable building category subsets. Base maps and built-up areas shapefile are from © OpenStreetMap contributors 2022. Distributed under the Open Data Commons Open Database License (ODbL) v1.0.**

| Hazard | Class | Weight |
|---|---|---|
| Water Depth (WD) | >0.71 m | 1 |
| | 0.70-0.51 m | 2 |
| | 0.50-0-31 m | 3 |
| | 0.30-0.21 m | 4 |
| | 0.20-0.00 m | 5 |
| Sink (SI) | Yes | 0 |
| | No | 1 |
| Normalized Imperviousness Surface Ratio (NISR) | 0.46-1.00 | 1 |
| | 0.46-0.23 | 2 |
| | 0.23-0.12 | 3 |
| | 0.12-0.05 | 4 |
| | 0.05-0.00 | 5 |

**Table 1: Weights related to Hazard classes for flood depth assigned to residential buildings, on sink map and on NISR classes.**

| Structural and non-structural feature | Class | Weight | Source |
|---|---|---|---|
| Construction Material Type (MT) | Other material | 1 | Italian National Institute for Statistics (2011 census, http://www.istat.it/) |
| | Concrete | 2 | |
| | Masonry | 3 | |
| Period of construction (PC) | Before 1919 | 1 | Italian National Institute for Statistics (2011 census, http://www.istat.it/) |
| | 1919-1945 | 2 | |
| | 1946-1960 | 3 | |
| | 1961-1970 | 4 | |
| | 1971-1980 | 5 | |
| | 1981-1990 | 6 | |
| | 1991-2000 | 7 | |
| | 2001-2005 | 8 | |
| | After 2006 | 9 | |

| | Bad | 1 | |
|---|---|---|---|
| Building Status (BS) | Average | 2 | Italian National Institute for Statistics (2011 census, http://www.istat.it/) |
| | Good | 3 | |
| | Very Good | 4 | |
| Building type (BT) | Detached/semi-detached houses (i.e., dwelling unit inhabited by a single household. Houses forming half of a semi-detached pair) | 1 | Open Street Maps dataset |
| | Residential (i.e., building used primarily for residential purposes) | 2 | |
| | Apartments (i.e., buildings arranged into individual dwellings, often on separate floors. May also have retail outlets on the ground floor) | 3 | |
| Building height (BH) | 1-5 | 1 | Italian Ministry of Environment's Geoportale Nazionale (http://wms.pcn.minambiente.it/ogc?map=/msogc/wfs/Edifici.map) |
| | 5-10 | 2 | |
| | 10-15 | 3 | |
| | 15-20 | 4 | |
| | 20-25 | 5 | |
| | 25-30 | 6 | |
| | 30-35 | 7 | |
| | 35-40 | 8 | |
| | 40-100 | 9 | |

**Table 2: Weights assigned to each structural and feature composing residential buildings, such as MT (i.e., Construction Material Type), PC (i.e., Period of construction), BS (i.e., Building Status), BH (i.e., Building Height), BT (i.e., Building Type).**

| Total Score | Vulnerability Class |
|---|---|
| 0-17 | Very High |
| 18-20 | High |
| 21-23 | Moderate |
| 24-27 | Low |
| 28-34 | Very Low |

**Table 3: Total Score and Vulnerability classes ('Very High', 'High', 'Moderate', 'Low', 'Very Low').**

| Vulnerability Class | Average economic unit value (€/m²) | Min economic unit value (€/m²) | Max economic unit value (€/m²) |
|---|---|---|---|
| Very Low | 3,944.41 | 2,862.50 | 10,050.00 |
| Low | 3,907.86 | 1,850.00 | 10,050.00 |
| Moderate | 4,033.10 | 2,112.50 | 10,050.00 |
| High | 4,065.08 | 2,112.50 | 6,200.00 |
| Very High | 4,138.20 | 2,112.50 | 5,450.00 |

**Table 4: The average, minimum and maximum economic unit value for each residential building structural vulnerability class.**

| | With basement | | | | No basement | | | |
|---|---|---|---|---|---|---|---|---|
| Subset | Flood 2014 [m] | 10 years scenario [m] | 100 years scenario [m] | 500 years scenario [m] | Flood 2014 [m] | 10 years scenario [m] | 100 years scenario [m] | 500 years scenario [m] |
| 1 | 0.03 | 0.11 | 0.21 | 0.08 | 0.01 | 0.13 | 0.12 | 0.09 |
| 2 | 0.02 | 0.13 | 0.20 | 0.11 | 0.01 | 0.07 | 0.07 | 0.06 |

**Table 5: Flood depths necessary for causing a DF equal to 0.1, according to the building type two subset.**

**Appendix**

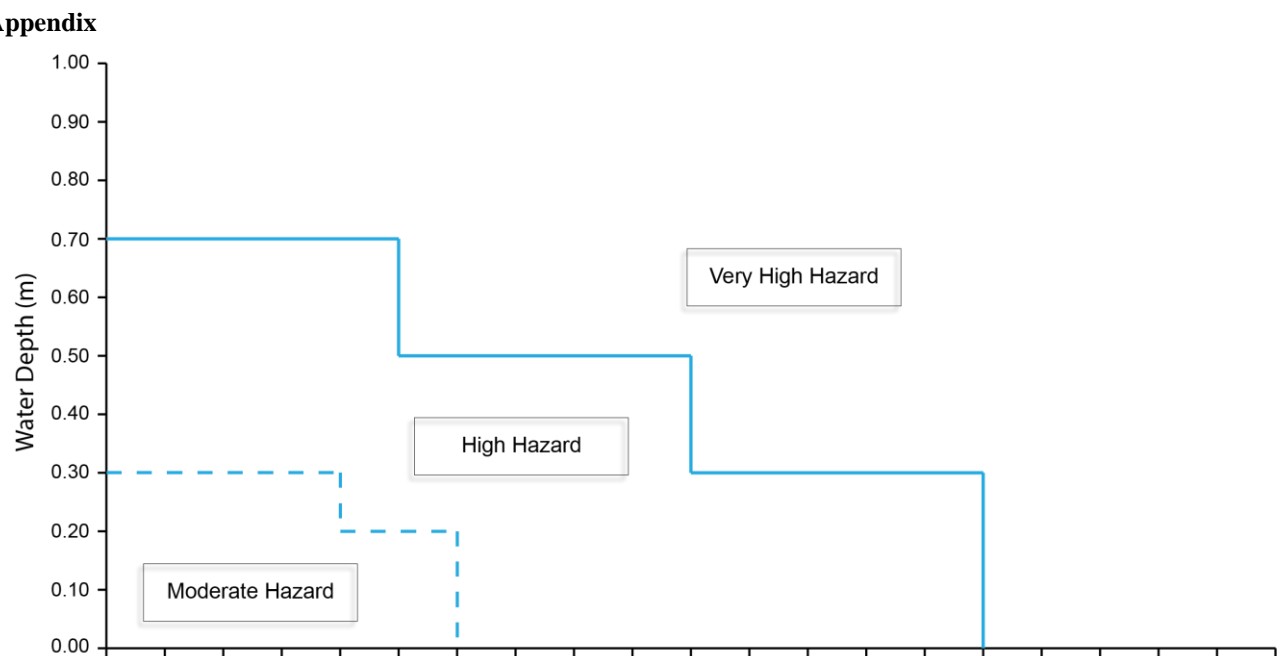

**Figure A1: Flood depth and velocity thresholds used as reference values for the flood hazard classification (adapted from Lombardy Region PTC).**

| Variable | Description | Unit of measurement | Range of values | Default Values | Input Values |
|---|---|---|---|---|---|
| he | Water depth outside the building | m | $\geq 0$ | [0;0.5 ] Incremental step: 0.01 | [0-20.4] Incremental step: 0.05 |
| h | Water depth inside the building (for each floor) | m | [0;1H] | h=f(he, GL) | h=f(he, GL) |
| v | Max velocity of the water perpendicularly to the building | m/s | $\geq 0$ | 0.5 | 0.5 |
| s | Sediment load | % on the water volume | [0;1] | 0.05 | 0.05 |
| d | Duration of the flood event | hours | >0 | 24 | 36 |

| | | | | | |
|---|---|---|---|---|---|
| q | Water quality (presence of pollutants) | - | 0: No  1: Yes | 1 | 1 |

**Table B1: Features parameters INSYDE model input data.**

| Variable | Description | Unit of measurement | Range of values | Default Values | Input Values |
|---|---|---|---|---|---|
| FA | Footprint area | m$^2$ | >0 | 100 | [11.6; 5400.7] |
| IA | Internal area | m$^2$ | >0 | 0.9*FA | 0.9*FA |
| BA | Basement area | m$^2$ | ≥0 | 0.5*FA | 0.5*FA |
| EP | External perimeter | m | >0 | 4*ξ | [14.1; 895.9] |
| IP | Internal perimeter | m | >0 | 2.5 EP | 2.5 EP |
| BP | Basement perimeter | m | >0 | 4*ξ | 4*ξ |
| NF | Number of floors | - | ≥1 | 2 | [1; 4] |
| IH | Interior floor height | m | >0 | 3.5 | 3.5 |
| BH | Basement height | m | >0 | 3.2 | 3.2 |
| GL | Ground floor level | m | [-IH;>0] | 0.1 | 0.1 |
| BL | Basement level | m | <0 | -GL-BH-0.3 | -GL-BH-0.3 |
| BT | Building type | - | 1: Detached house  2: Semi-detached house  3: Apartment | 1 | [1; 2; 3] |
| BS | Building structure | - | 1:reinforced concrete  2: Masonry | 2 | [1; 2] |
| FL | Finishing level (i.e., building quality) | - | 0.8: low  1:medium  1.2: high | 1.2 | 1.2 |
| LM | Level of maintenance | - | 0.9: low  1: medium  1.1: high | 1.1 | [0.9; 1; 1.1] |
| YY | Year of construction | - | ≥0 | 1994 | [1919; 2006] |
| PD | Heating system distribution | - | 1: centralized  2: distributed | 1 if YY≤ 1990  2 otherwise | 1 if YY≤ 1990  2 otherwise |
| PT | Heating system type | - | 1: radiator  2: pavement | 2 if YY≥ 2000 and FL>1  1 otherwise | 2 if YY≥ 2000 and FL>1  1 otherwise |

**Table B2: Building characteristics parameters INSYDE model input data.**

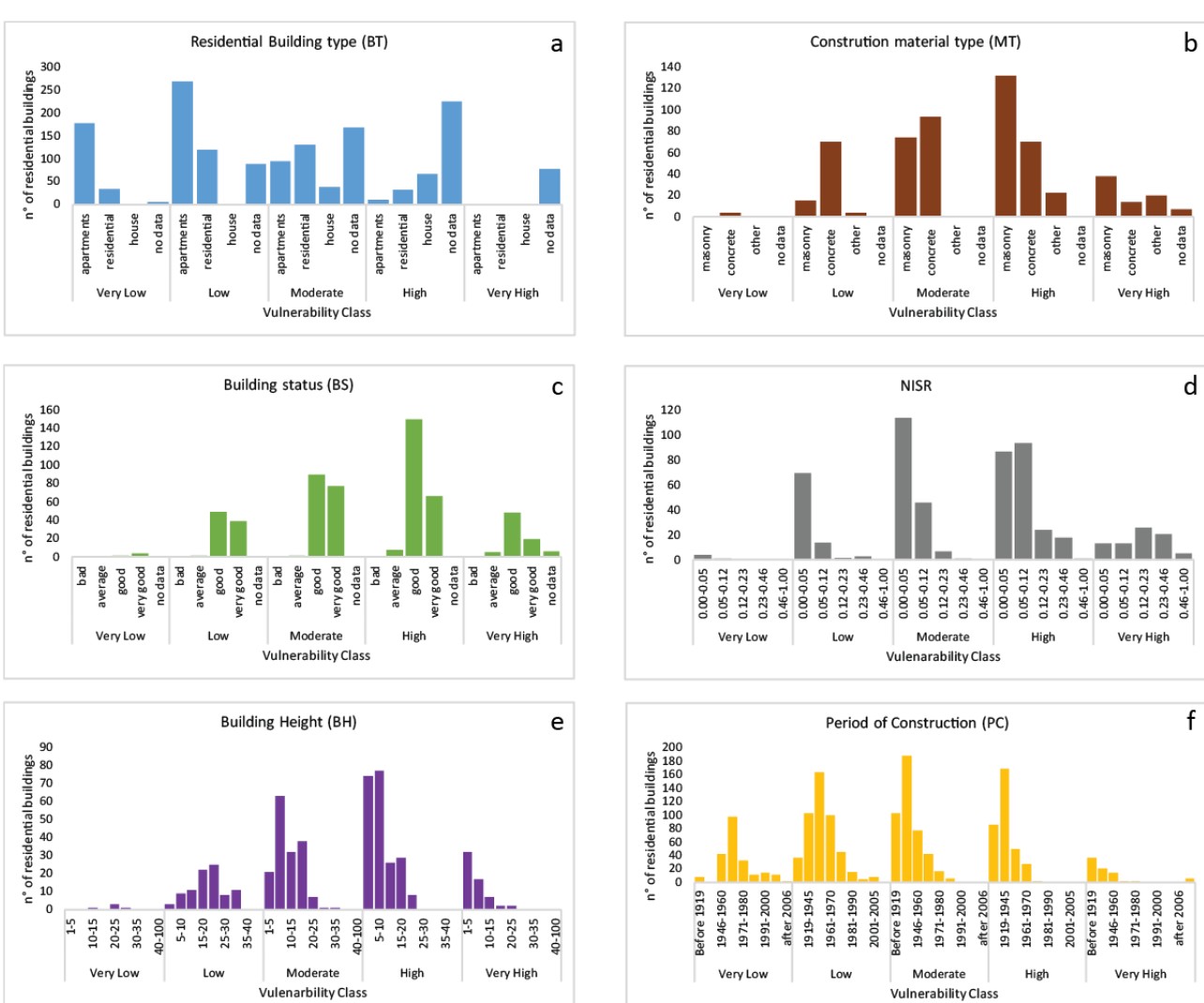

**Figure C1: Structural and non-structural building characteristics vulnerability class distribution: Residential Buildings Type (a), Construction Material Type (b), Building Status (c), Normalized Imperviousness Surface Ratio) (d), Building Height (e), Period of Construction (f).**