# Peer review of "Building-scale flood loss estimation through vulnerability pattern characterization: application to an urban flood in Milano, Italy"

_EGUsphere, 2022_

## Referee Comment (RC1)

**REVIEWER'S REPORT**

I have now read the article "Building-scale flood loss estimation through enhanced vulnerability pattern characterization: application to an urban flood in Milano, Italy". In my opinion, the article needs a major revision before it can be accepted for publication. The main issues related to this article are:

**GENERAL COMMENTS**

-**Lack of focus**: the focus of the paper is weak. It is not clear if the authors plan to focus on loss estimation, enhance vulnerability assessment, focus on the use of indicators or the use of vulnerability/damage curves and why the study is important and for who.

-**Theoretical background:** It is not clear what the authors understand with the term "vulnerability". Generally, the reference to the theoretical background is very weak. I see the following connection:

Source+pathway estimation= flood modelling (susceptibility mapping), Receptor = exposure, Consequence = loss

But then what is vulnerability? Is it receptor+consequence? Is it the monetary loss? Or a pre-existing condition based on the building characteristics? It is not clear. And what about risk? Throughout the manuscript, the term vulnerability is used in different ways.

-**Clarity of the description of the methodology:** The methodology is not clear. A methodological workflow figure would certainly help. Additional and detailed information is needed regarding the choice of indicators, their scoring and weighting. Discussion and justification are needed in issues such as e.g. the use of absolute monetary loss.

-Important **publications on the topic are not mentioned** and cited. A **short literature** review and the presentation of existing gaps that are going to be filled are absolutely necessary. The authors have to show what has been available until now, and what are the gaps that they are filling with their research. This is not clear in the manuscript.

- A connection between **results and possible end-users** and their needs is entirely missing.

-The results are described but **not adequately interpreted.**

-The **limitations** part is very short and weak. What were the assumptions and limitations of the study and (based on them) what could be the **future research development**?

-Most of the **figures** are of poor quality and the legends cannot be read. The choice of colour for different categories is not representative.

-The **title** has to be reconsidered after the completion of the review. What do the authors mean with "enhanced" vulnerability pattern characterization?

SPECIFIC COMMENTS:

*Comments on the text:*

Line 37-40: Reference is missing (Vogel and O'Brian, 2004).

Line 46: "how vulnerability is generated". Is vulnerability generated? Or is it an existing condition? Please check the general issues above regarding the theoretical background of the study, definition and understanding of vulnerability.

Line 58: the authors have to refer at this point to existing publications and authors that have worked intensively with indicator-based methods for floods or flash floods and similar phenomena (see references below):

Balica, S. F., Douben, N. & Wright, N. G. Flood vulnerability indices at varying spatial scales. *Water Sciences and Technology* **60.10**,2571–2580, https://doi.org/10.2166/wst.2009.183 (2009).

Kappes, M.S.; Papathoma-Köhle, M. & M. Keiler, 2012. Assessing physical vulnerability for multi-hazards using an indicator-based methodology. Applied Geography, 32, 2, 577-590. https://doi.org/10.1016/j.apgeog.2011.07.002

Papathoma-Köhle, M.; Schlögl, M.; Fuchs, S. 2019. Vulnerability indicators for natural hazards: an innovative selection and weighting approach. *Scientific Reports,* 9, 15026. https://doi.org/10.1038/s41598-019-50257-2

Line 62: Akbas (2009): please check again. Akbas does NOT use an indicator-based approach he has developed a vulnerability curve.

Lines 76-77: "source, pathway and receptor refer to the physical process". Why does "receptor" refer to the physical process? Later on, it is stated that the receptor "refers to the entities that may be damaged by the hazards". Please revise.

Line 179: why weight "0" for buildings with basement and not the other way round? It seems that the weights are low for buildings that are more vulnerable and this needs to be clarified and justified. This is obvious also in Table 1.

Table 1: The same comment as above. Why new buildings have more weight than old ones? They should be less vulnerable or not? Who decides what is "bad"? Who decides the building status? What are the criteria? Is this expert judgement and who are the experts? The authors themselves? A panel of external experts (number and background)? Please indicate metrics. The height of the building is in meters, right?

Moreover, I think that with the word "weight" the authors mean "score". The "weight" should show the importance and hierarchy of the different indicators. For example, if the material of the building is more important than the height then the weight of the material should be higher. The score would show the individual score that would be assigned to a building for a particular indicator (e.g. for the indicator "building material" the score for wooden buildings is 1). Furthermore, a general discussion about the weight is missing. What kind of weighting is there in the literature? Is "equal weighting" acceptable? How can we acquire weights? (e.g.

statistical or participatory methods). The authors seem to follow an indicator-based approach. What are the weights of the indicators? If the authors have decided not to use any, why not?

Lines 175-177: So vulnerability classes equal a relative market value? Once more it is not clear what vulnerability is for the authors. Is the market value better to use in the assessment than the reconstruction value and why? The market value can be influenced by many factors such as view, the safety of the neighbourhood, access to infrastructure, touristic attraction, etc.

Lines 191-194: is only the scenario of 2014 taken into consideration or also the 10, 100, 500-year flood? Not clear...

Line 202: reference to a different method (curves) is rather confusing but using indicators to inform damage curves is very interesting. On the positive side, if this is really the case, I propose to emphasize it more and make it clear from the beginning of the paper.

Line 214-215: more info and detail are required. In most vulnerability or damage curves the "degree of loss" is used. What exactly is the case in the present study and why?

Line 216: "by building footprint"? what happens with multi-storey buildings?

Lines 245-262: I find this part very difficult to follow. Again, here, a methodological workflow would be helpful.

Line 253: Who decides these classes (High, Low, Medium etc.) and under what criteria? Who is the receptor of these results (and the potential end-user of the study?) What are their needs and how are these covered by the results?

Lines 271-273: How do the authors interpret these results?

Lines 280-283: How do the authors interpret these results?

Line 284: "buildings falling within 'Other roads and associated land'" How do buildings belong to a category named "roads".

Line 290-291: "making a distinction between elements with and without basement" why is this distinction necessary?

Line 371: In which way was the understanding of vulnerability improved?

Line 377-378: The authors should elaborate more here. Reference to existing curves is also missing. This is a good source of the recent advances in the field:

Fuchs, S.; Keiler, M.; Ortlepp, R.; Schinke, R.; Papathoma-Köhle, M. 2019. Recent advances in vulnerability assessment for the built environment exposed to torrential hazards: Challenges and the way forward. *Journal of Hydrology,* 575: 587-595. https://doi.org/10.1016/j.jhydrol.2019.05.067

Line 377-383: The limitations part of the paper is very short. All the assumptions and limitations have to be acknowledged.

Conclusions: there is a focus on vulnerability here whereas the paper seemed to have as an aim the calculation of loss. Furthermore, there is no discussion about possible uses and end-users of the presented research.

*Comments on the Figures:*

Figure 5: Attention: in the map legend it is NISR instead of NIRS.

Figure 7: Why this choice of colours? For example, high buildings should be less vulnerable and experience a lower degree of loss (red colour) and the low buildings that should be more vulnerable are green.

Figure 7d: what is the difference between apartment, residential and attached/semi-detached? Are they not all residential?

Figure 7f: Building values are uniform? Is it because they are the market values? So, they have nothing to do with the physical vulnerability in my opinion.

Figure 8: Similar comment about the choice of colours for buildings with and without a basement.

Figure 8: what are "other buildings" and "other vulnerability class"?

Figure 9: The difference between the three maps is minimal. Is it because it is only a smaller part of a larger inundation map? How do the authors explain this similarity?

Figures 10, 11: damage curves with absolute damage make no sense to me. What kind of information do they deliver that can be useful? Give more information about the making of the curves. Which regression model did the authors use? (Weibull distribution, logistic?). How do the authors interpret the fact that after a point the curves are steady? Why do they not extend beyond a specific water depth? What is the role of the number of floors? And most importantly: what are the potential uses and end-users of these curves?

---

## Author Comment (AC1)

**Comment on egusphere-2022-225**

**Answer to Anonymous Referee #1**

I have now read the article "Building-scale flood loss estimation through enhanced vulnerability pattern characterization: application to an urban flood in Milano, Italy". In my opinion, the article needs a major revision before it can be accepted for publication. The main issues related to this article are:

*We thank the anonymous reviewer for his/her careful reading of the manuscript and his/her constructive remarks. We have taken the comments on board to improve and clarify the manuscript. The vast majority of those have resulted in an addition to the text or to a change. Please find below a detailed point-by-point response to all comments (Referee's comments in black, our replies in red).*
*Major general changes:*
*• Modify the abstract to reflect clearer the scope of the paper;*
*• Added a clearer 'Background' and 'Aims' sections;*
*• Provided more detail on the Methodology throughout the manuscript;*
*• Ensuring a consistent results interpretation in the Discussion;*
*• Enhanced the figures.*

GENERAL COMMENTS

-**Lack of focus**: the focus of the paper is weak. It is not clear if the authors plan to focus on loss estimation, enhance vulnerability assessment, focus on the use of indicators or the use of vulnerability/damage curves and why the study is important and for who.

**Reply:** *Following the Referee advice the authors have made paper aims clearer in the Introduction Section and highlighting the potential of the results obtained to support decision makers, as planners and policy makers, in improving their investment strategies for the mitigation and the reduction of flood damages in the Discussion.*

*'The aim of this paper is to improve structural vulnerability assessment for a flood prone area considering vulnerability to be a composition of the hazard intensity and inherent characteristics of elements-at-risk and their surroundings as sources of information to define the more susceptible and exposed residential buildings to inform depth-damage curve and calculate potential losses.'*

*'Here vulnerability assesses the potential susceptibility consequences of the hazard to the exposed residential buildings to: 1) address the necessity of an integrated qualitative and quantitative approach to define flood physical (structural) vulnerability; 2) to accurately re-assessed past event flood characteristics; 3) to bridge the gap of vulnerability spatialization using spatially distributed information to obtain vulnerability mapping and related potential damage suitable for giving indications on where and how to reduce risks at local level; 4) to bridge the lack of reliable input data in particular regarding the properties of the element at risks (i.e., the exposure and susceptibility factors) that also contribute to their vulnerability at building level; 5) improve the exposure analysis integrating the building surrounding as vulnerability indicator (i.e., morphological features and land use).'*

-**Theoretical background:** It is not clear what the authors understand with the term

"vulnerability". Generally, the reference to the theoretical background is very weak. I see the following connection:

Source+pathway estimation= flood modelling (susceptibility mapping), Receptor =

exposure, Consequence = loss

But then what is vulnerability? Is it receptor+consequence? Is it the monetary loss? Or a pre-existing condition based on the building characteristics? It is not clear. And what about risk? Throughout the manuscript, the term vulnerability is used in different ways.

*Reply: 'Source' and 'Pathway' represent the flood hazard. 'Source' is determined by the probability (p) of flood events with a certain magnitude and other features, the 'Pathway' can be described by the inland inundation with various attributes for flood control. 'Receptor' and '(negative) Consequence' state the vulnerability, whereas 'receptor' specifies the exposure and the susceptibility , negative consequences refer to loss estimation. Following the Referee suggestions the Authors have rearranged the Introduction trying to make vulnerability definition clearer according to the study aims and methodologies used. We also tried to make it clearer and more uniform throughout the entire manuscript as follows:*

*'We focus here on the structural dimension of vulnerability of flood-prone areas that has proven to be key to analysing flood risk. Since the structural vulnerability is defined as the potential of a particular class of buildings or infrastructure facilities to be affected or damaged under a given flood intensity (Faella and Nigro, 2003), damage by flood hazard depends on the vulnerability of exposed buildings (Schanze J., 2006; Merz et al., 2010).'*

*'The aim of this paper is to improve structural vulnerability assessment for a flood prone area considering vulnerability to be a composition of the hazard intensity and inherent characteristics of elements-at-risk and their surroundings as sources of information to define the more susceptible and exposed residential buildings to inform depth-damage curve and calculate potential losses.'*

*'Here vulnerability assesses the potential susceptibility consequences of the hazard to the exposed residential buildings to: 1) address the necessity of an integrated qualitative and quantitative approach to define flood physical (structural) vulnerability; 2) to accurately re-assessed past event flood characteristics; 3) to bridge the gap of vulnerability spatialization using spatially distributed information to obtain vulnerability mapping and related potential damage suitable for giving indications on where and how to reduce risks at local level; 4) to bridge the lack of reliable input data in particular regarding the properties of the element at risks (i.e., the exposure and susceptibility factors) that also contribute to their vulnerability at building level; 5) improve the exposure analysis integrating the building surrounding as vulnerability indicator (i.e., morphological features and land use)'*

-**Clarity of the description of the methodology:** The methodology is not clear. A methodological workflow figure would certainly help. Additional and detailed information is needed regarding the choice of indicators, their scoring and weighting. Discussion and justification are needed in issues such as e.g. the use of absolute monetary loss.

*Reply: The Methodology section has been improved adding more details and merging Figure 2 and Figure 3 to make the methodology framework clearer. Furthermore additional and detailed information have been also added as follow:*

*'Age and maintenance are also indications for the current state of the building. Moreover, an estimation of elements-at-risk costs is fundamental to express losses in economic terms. The heuristic approach is based on a simple equal weights assignment procedure (Taramelli et al., 2015). The weights assignment have been given by authors based on an intensive literature review and on data availability. The indicators are identified as significant and suitable vulnerability indicators in*

*urban areas representative of three components, i.e., flood hazard intensity, effect of the surrounding environment and building characteristics.'*

*'Here the building economic unit value has been used for estimating the exposed assets in terms of monetary exposure of the residential buildings in the flooded area assuming solely the structure value excluding the content value. For buildings falling in most vulnerable classes (i.e., class 1 and 2, 'Very High' and 'High') a distinction was made between elements with and without basement (Arrighi et al., 2020; Molinari et al., 2020) assigning a weight of 0 to the building with basement and 1 to the building without basement (McBean et al., 1988; Crigg and Helweg, 1975). As previously mentioned, the assignment of weights refers to the building's response capacity against flood hazard, meaning the capability or incapability of an object to resist the flood impact. Hence, lower values correspond to lower response capacity (high vulnerable buildings), whereas higher values correspond to higher response capacity (low vulnerable buildings) (Taramelli et al., 2015).'*

*'INSYDE model can estimate relative damage (i.e., percentage estimation of losses with respect to the total value of the building) and absolute damage, the latter considers the unit prices of cost of damage. Here we decide to express the damage in absolute terms to give the monetary measure using the cost per unit of measure (e.g., square meter). Here, the INSYDE model was applied deterministically (i.e., without considering any source of uncertainty). To calculate the damage we combine the exposure and vulnerability data described above with the 2014 flood depth and considering three existing diverse flood depth scenarios assuming return periods of 10, 100 and 500 years derived from 1-D and 2-D hydraulic modelling designed by the Municipality of Milano in 2019 for the Governmental Territorial Plan and the Flood Risk Management Plan.'*

-Important **publications on the topic are not mentioned** and cited. A **short literature** review and the presentation of existing gaps that are going to be filled are absolutely necessary. The authors have to show what has been available until now, and what are the gaps that they are filling with their research. This is not clear in the manuscript.

*Reply: As stated before following the Referee suggestion the authors have rearranged the Introduction, citing important publications on the topic and highlighting some of the existing gaps coming up from the literature review, to 1) address the necessity of an integrated qualitative and quantitative approach to define flood physical (structural) vulnerability; 2) to accurately re-assessed past event flood characteristics; 3) to bridge the gap of vulnerability spatialization using spatially distributed information to obtain vulnerability mapping and related potential damage at land use and building levels suitable for giving indications on where and how to reduce risks at local level; 4) to bridge the lack of reliable input data in particular regarding the properties of the element at risks (i.e., the exposure and susceptibility factors) that also contribute to their vulnerability; 5) the integration of the building surrounding as vulnerability indicator (i.e., morphological features and land use). We have argued better about this in the Introduction as follows:*

*'As a result of neglected attention to disaster risk impacts in the past, it is not easy or even possible to portray the spatial and temporal patterns of flood damage and losses with reasonable precision. This makes measurement of progress in reducing the disaster risk difficult if not impractical. Therefore, some authors stressed the need for the use of empirical data from past events to provide powerful analytical tools (Vamvatsikos et al., 2010; Apel et al., 2008). The ideal solution would be a model that is simple, as the empirical ones, but that include quantitative and qualitative explicative parameters in damage assessment, as the synthetic ones to capture the full complexity of buildings vulnerability (Lal et al., 2012; Morelli et al., 2021; Balica et al., 2009; Schanze J., 2006). The aim of this paper is to improve structural vulnerability assessment for a flood prone area considering vulnerability to be a composition of the hazard intensity and inherent characteristics of elements-at-*

*risk and their surroundings as sources of information to define the more susceptible and exposed residential buildings to inform depth-damage curve and calculate potential losses. To better understand potentially damaging natural processes interacting with elements at risk the work is structured according to the modified Source-Pathway-Receptor-Consequence (SPRC) conceptual model (Fleming, 2002). The SPRC model relies on a simple causal chain commonly adopted to understand the flood risk system and the link among processes (Schanze J., 2006). The SPRC model consists of the hazard origin identification as to an event transmitted through a pathway to a receptor with possible negative effects for the receptor depending on their vulnerability and their exposure intensity considering events of different magnitudes (Hallegatte et al., 2013; Taramelli et al., 2015). Here vulnerability assesses the potential susceptibility consequences of the hazard to the exposed residential buildings to: 1) address the necessity of an integrated qualitative and quantitative approach to define flood physical (structural) vulnerability; 2) to accurately re-assessed past event flood characteristics; 3) to bridge the gap of vulnerability spatialization using spatially distributed information to obtain vulnerability mapping and related potential damage suitable for giving indications on where and how to reduce risks at local level; 4) to bridge the lack of reliable input data in particular regarding the properties of the element at risks (i.e., the exposure and susceptibility factors) that also contribute to their vulnerability at building level; 5) improve the exposure analysis integrating the building surrounding as vulnerability indicator (i.e., morphological features and land use).'*

- A connection between **results and possible end-users** and their needs is entirely missing.

*Reply:* *The authors provided a new subparagraph 5.3 Advantages, limitations and future development where the main study's findings have been matched with potential end-users and their needs.*

*'Lombardy, Italy's economic engine, is particularly vulnerable to flood hazard risk, the consequences of which are likely to affect the national growth and stability. A better understanding of vulnerability patterns is important for public budgeting, as well as private resilience choices. The study presents several advantages to support decision makers, as planners and policy makers, in improving their investment strategies for the mitigation and the reduction of flood damages, but also to improve the currently low flood insurance coverage exploring new insurance model trials to sustain insurance system for residential properties and (Gizzi et al., 2016). The SPRC model is a valuable tool to support decision makers in better understanding vulnerability drivers, complex processes and their interrelations acting as a guide for interventions, priority measures and resources allocation either in the pre or post- event phases. Examples of pre-flood applications of the outputs may include the development of flood hazard mitigation strategies that outline policies and programs for reducing flood losses, including nature-based solutions or the use of the obtained AAL as an input of a cost-benefits analysis of prevention and mitigation measures. Thereby examples of post-event applications of the outputs may include the application of land use planning principles and practices, the allocation of resources for flood-resilient buildings interventions. Therefore, the use of an indicator-based methodology to inform the depth-damage curve allows users to add indicators according to their needs, priorities and data availability. Herein, besides using only one characteristic of the building, valuable vulnerability building indicators based on structural and non-structural attributes and surrounding characteristics that contribute to the vulnerability of the elements at risk are used to inform damage curves identifying the hotspots. The use of spatially distributed information to obtain vulnerability mapping and related potential damage at land use and building levels gives useful indications on elements that will most likely experience the impact of a flood event and consequently where and how specific intervention of protection and maintenance or particular insurance against flood damage could be required. Moreover, the use of the environmental features*

*characteristics to define building surroundings in the susceptibility assessment play an important role in capturing the totality of the landscape elements that influence the vulnerability patterns amplifying or diminishing vulnerability of elements at risk.'*

-The results are described but **not adequately interpreted.**

*Reply: The results have been described in the Results section and interpreted mostly in the Discussion section. Though we have adequately rearranged some results interpretation to strengthen the results robustness as follows:*

*'Overall AAL represents the full range of hazard magnitudes offering a more complete picture of monetary impacts should expect to incur over time. Thereby exposed assets as a function of water level were then translated into absolute damage. To calculate the damage we combine the exposure and vulnerability data with the 2014 flood water depth assuming return periods of 500 , 100 and 10 years. Flood depths for different return periods show small differences, due to the relatively flat area where the floodwaters can spread. Localised spots with flood depths larger than 4 metres occur in depressions being filled by the floodwaters (e.g., metro stations, road underpasses). 'Continuous urban fabric (S.L.:> 80%)' class is mostly affected by low flood depth (less than 0.2-0.5 m) for the 2014 food and 10 years return period than the other two scenarios (Figure 9). Nevertheless, at mesoscale the 'Continuous urban fabric' Urban Atlas 2018 land-use class with the occurrence of at least 80 % of soil sealing shows higher absolute damage values within the first meter according to existing literature the damage factor on built up areas is different for diverse types of land use (Huizinga et al., 2017; Gabriels et al., 2022), but tends to be steeper within the first meter. Residential buildings falling in the 'Roads and associated land' class likewise show great absolute damage values. Roads accounted for the increase in impervious cover giving a significant impact on natural water systems (Figs. 11 and 12). Whereas less values of absolute damage are measured for those building located close to 'Green urban areas' is noteworthy (Figs. 11 and 12). At microscale observing the depth–damage functions for the residential sector the shape of the damage for the 2014 flood is steeper to 0.2 m water depth than other functions. Nearly maximum damage occurs when water depths exceed 0.5 m probably because observing the 'Very High' and 'High' vulnerable building distribution against water depth classes is evident that mostly vulnerable buildings fall within 0.26-0.50 m class.'*

-The **limitations** part is very short and weak. What were the assumptions and limitations of the study and (based on them) what could be the **future research development**?

*Reply: The authors provided a new subparagraph 5.3 Advantages, limitations and future development where main study's limitations have been described in greater detail and used as starting point for future research development proposals as follows:*

*'Nevertheless, some limitations can be noted. Considerable amount of data at local scale is needed, however it is not always available to the local authorities or cannot be accessible or easily collected. In Italy the current national and regional databases are often of insufficient quality to support a robust analysis. One of the three main elements to be correlated among hazard, vulnerability, and losses is often missing or too hazy to make an appropriate comparison with scientific findings (Molinari et al., 2014). Here effects of hazard interactions on the buildings susceptibility and exposure is hazard-intensity specific based on 2014 flood event severity, however to provide more powerful analytical tools more empirical data should be included. Several studies have shown that estimations based on stage-damage functions may be very uncertain since water depth and building use only represent a fraction of the whole data variance (Merz et al. 2004). Therefore, the lack of loss data or scattered information on the damage suffered by buildings do not allow a proper*

*comparative assessment between past damage flood events data to those findings obtained by modelled depth-damage functions. As well, better information on past events and damage occurrence and amount would provide more information regarding the building response capacity and would improve heuristic approach and susceptibility assessment. Thus, the scoring procedure in this study has been done on the basis of literature having many associated uncertainties. A conceivable future development would be to base the scoring procedure on documentation of other available past events occurred in the study area or of future events collecting and rehabilitating the scattered data on the economic impact of past flood events, including the indemnities paid to insurance policy holders, compensations paid for uninsured residential flood damage and state aid provided to economic entities to foster recovery. As for the latter point, available products and services of the European Copernicus Earth Observation program can be a valid support to bridge this gap. Specifically, Copernicus Emergency Management Service that could provide information about the damage grade, its spatial distribution and extent (i.e., grading map) derived from images acquired in the aftermath of the flood event. Furthermore, the approach could certainly be broadened to commercial buildings and critical facilities (e.g., school, hospitals, public buildings etc.) enlarging the dataset and the study area.'*

-Most of the **figures** are of poor quality and the legends cannot be read. The choice of colour for different categories is not representative.

**Reply:** *The figures in the manuscript have low resolution to limit file size. High resolution figures are provided.*

-The **title** has to be reconsidered after the completion of the review. What do the authors mean with "enhanced" vulnerability pattern characterization?

**Reply:** *The authors with the term "enhanced" vulnerability pattern characterization mean the improvement of vulnerability parametrization i) re-assessing accurately past flood characteristics, ii) using spatially distributed information at land use and building level to define elements at risk exposure, susceptibility and response capacity to hazardous event, iii) improving input data quality to implement site-specific depth-damage curves to depict the spatial patterns of flood damage and losses with reasonable precision. However, for more clarity as suggested by the Referee the title has been changed to 'Building-scale flood loss estimation through enhanced vulnerability pattern characterization: application to an urban flood in Milano, Italy'.*

SPECIFIC COMMENTS:
*Comments on the text:*
Line 37-40: Reference is missing (Vogel and O'Brian, 2004).

**Reply:** *Reference has been included.*

Line 46: "how vulnerability is generated". Is vulnerability generated? Or is it an existing condition? Please check the general issues above regarding the theoretical background of the study, definition and understanding of vulnerability.

**Reply:** *The sentence has been rephrased for clarity as follows:*
*'Thus, it is essential to address the drivers of vulnerability , how it increases, and how it is distributed to effectively manage risk.'*

Line 58: the authors have to refer at this point to existing publications and authors that have worked intensively with indicator-based methods for floods or flash floods and similar phenomena (see references below):

Balica, S. F., Douben, N. & Wright, N. G. Flood vulnerability indices at varying spatial scales. *Water Sciences and Technology* **60.10**,2571–2580, https://doi.org/10.2166/wst.2009.183 (2009).

Kappes, M.S.; Papathoma-Köhle, M. & M. Keiler, 2012. Assessing physical vulnerability for multi-hazards using an indicator-based methodology. Applied Geography, 32, 2, 577-590. https://doi.org/10.1016/j.apgeog.2011.07.002

Papathoma-Köhle, M.; Schlögl, M.; Fuchs, S. 2019. Vulnerability indicators for natural hazards: an innovative selection and weighting approach. *Scientific Reports,* 9, 15026. https://doi.org/10.1038/s41598-019-50257-2

Line 62: Akbas (2009): please check again. Akbas does NOT use an indicator-based approach he has developed a vulnerability curve.

**Reply:** *Authors included the suggested references in the Introduction rearranging the part related to the indicator-based methods to strengthen the literature review. We also correct the reference related to Akabas, 2009 as follows:*

*'Akbas et al., (2009) proposed a specific physical vulnerability curve introducing the concept of probabilistic damage functions and appropriate definition of relevant damage states, assessing temporal and spatial impact probability uncertainties.'*

Lines 76-77: "source, pathway and receptor refer to the physical process". Why does receptor" refer to the physical process? Later on, it is stated that the receptor "refers to the entities that may be damaged by the hazards". Please revise.

**Reply:** *Receptor refers to a physical process in the way it includes elements at risk that may be affected by physical impacts (see Shanze, 2006).*

*Schanze, J.: Flood Risk Management – A Basic Framework, in: Flood Risk Management: Hazards, Vulnerability and Mitigation Measures, Springer Netherlands, Dordrecht, 1–20, https://doi.org/10.1007/978-1-4020-4598-1_1, 2007.*

Line 179: why weight "0" for buildings with basement and not the other way round? It seems that the weights are low for buildings that are more vulnerable and this needs to be clarified and justified. This is obvious also in Table 1.

**Reply:** *The assignment of weights refers to the building's response capacity against flood hazard, meaning the capability or incapability of an object to resist the flood impact. Hence, lower values correspond to lower response capacity (high vulnerable buildings), whereas higher values correspond to higher response capacity (low vulnerable buildings).*

Table 1: The same comment as above. Why new buildings have more weight than old ones? They should be less vulnerable or not? Who decides what is "bad"? Who decides the building status? What are the criteria? Is this expert judgement and who are the experts? The authors themselves? A panel of external experts (number and background)? Please indicate metrics. The height of the building is in meters, right? Moreover, I think that with the word "weight" the authors mean "score". The "weight" should show the importance and hierarchy of the different indicators. For example, if the material of the building is more important than the height then the weight of the material should be higher. The score would show the individual score that would be assigned to a building for a particular indicator (e.g. for the indicator "building material" the score for wooden buildings is 1).

Furthermore, a general discussion about the weight is missing. What kind of weighting is there in the literature? Is "equal weighting" acceptable? How can we acquire weights? (e.g. statistical or participatory methods). The authors seem to follow an indicator-based approach. What are the weights of the indicators? If the authors have decided not to use any, why not?

*Reply: As stated above the assignment of weights refers to the building's response capacity against flood hazard, meaning the capability or incapability of an object to resist the flood impact. Hence, lower values correspond to lower response capacity (high vulnerable buildings), whereas higher values correspond to higher response capacity (low vulnerable buildings) (Table 2). The weights assignment have been given by authors based on an intensive literature review and on data availability. The indicators are identified as significant and suitable vulnerability indicators in urban areas representative of three components, i.e., flood hazard intensity, effect of the surrounding environment and building characteristics. A final total score for In the heuristic approach is calculated by summing up all the weights assigned to each indicator composing a residential building. For indicators like Construction Material, Period of Construction, Building Status ranges like 1-9 or 1-4 were chosen, following the study of Corradi et al. (2015). In virtue of the possible methods (statistical assessments such as Principal Component Analysis, factor analysis, or participatory approaches like Analytic Hierarchy Process or Budget Allocation Process) an approach using equal weights by Taramelli et al. (2015) was selected, which represents the most adopted approach in literature (Beccari 2016; Papathoma-Köhle 2019). These indicators are identified as significant and suitable vulnerability indicators in urban areas representative of three components, i.e., flood hazard intensity, effect of the surrounding environment and building characteristics. A final total score for the heuristic approach is calculated by summing up all the weights assigned to each indicator composing a residential building, and classifying them in 5 categories, from 'Very High', to 'Very Low' vulnerability (Rincon et al. 2018, Gatti, 2020) using the Jenks Natural Breaks algorithm. With natural breaks classification (Jenks) Natural Breaks Jenks, classes are based on natural groupings inherent in the data. Class breaks are created in a way that best groups similar values together and maximizes the differences between classes. In addition, a different consideration was made to buildings with and without a basement. Here the aim was to set up the main vulnerability indicators based on the SPRC chain based on hazard intensity, building surrounding and buildings intrinsic characteristics. Overall, because of the method selected, the model output is quite sensitive to assumed levels of parameters. A robust sensitivity analysis would be a further step.*

Beccari, B.: A comparative analysis of disaster risk, vulnerability and resilience composite indicators." PLoS currents, 8, https://doi.org/10.1371/currents.dis.453df025e34b682e9737f95070f9b970, 2016.

Corradi, J., Salvucci, G., and Vitale, V.: Analisi della vulnerabilità sismica dell'edificato italiano : tra demografia e " domografia " una proposta metodologica innovativa, 1–22, 2015.

Gatti, I.: Disaster risk assessment for urban areas : A GIS flood risk analysis for Luján City (Argentina), M.Sc. thesis, The University of Tokyo, Japan, 75pp, 2020.

Papathoma-Köhle M, Cristofari G, Wenk M and Fuchs S: The importance of indicator weights for vulnerability indices and implications for decision making in disaster management, International journal of disaster risk reduction, 36,101103, https://doi.org/10.1016/j.ijdrr.2019.101103, 2019.

Rincón, D., Khan, U.T. and Armenakis, C.: Flood risk mapping using GIS and multi-criteria analysis: A greater Toronto area case study, Geosciences, 8, 275, https://doi.org/10.3390/geosciences8080275, 2018.

Taramelli, A., Valentini, E., and Sterlacchini, S.: A GIS-based approach for hurricane hazard and vulnerability assessment in the Cayman Islands, Ocean Coast. Manag., 108, 116–130, https://doi.org/10.1016/j.ocecoaman.2014.07.021, 2015.

Lines 175-177: So vulnerability classes equal a relative market value? Once more it is not clear what vulnerability is for the authors. Is the market value better to use in the assessment than the reconstruction value and why? The market value can be influenced by many factors such as view, the safety of the neighbourhood, access to infrastructure, touristic attraction, etc.

*Reply: Here market values have been used for estimating the exposed assets in terms of monetary exposure of the residential buildings in the flooded area. Whereas, in the definition of the vulnerability curves using the INSYDE model damage in absolute terms is calculated considering the replacement or reconstruction value of damaged components.*

Lines 191-194: is only the scenario of 2014 taken into consideration or also the 10, 100, 500-year flood? Not clear...

*Reply: To calculate the damage we consider four scenarios, the 2014 flood scenario, and the 10, 100, 500-year flood.*

Line 202: reference to a different method (curves) is rather confusing but using indicators to inform damage curves is very interesting. On the positive side, if this is really the case, I propose to emphasize it more and make it clear from the beginning of the paper.

*Reply: As suggested by the Referee we emphasized this latter point through the manuscript stressing this purpose from the beginning of the paper, in the Abstract and in the Introduction.*

Line 214-215: more info and detail are required. In most vulnerability or damage curves the "degree of loss" is used. What exactly is the case in the present study and why?

*Reply: In this work, the degree of loss, intended as the ratio between damage costs and the value of each individual element at risk (Fuchs et al., 2014), has not been calculated. Here Damage mechanisms considered in INSYDE adopt probabilistic and deterministic functions. Deterministic functions are adopted when the damage mechanism is well understood based on literature and author's opinion, and when the uncertainty of variability between parameters is small. Probabilistic functions are adopted when there is uncertainty of influence of parameters in the damage mechanism and uncertainty of thresholds for damage occurrence (Dottori et al., 2016).*

Fuchs, S.; Keiler, M.; Ortlepp, R.; Schinke, R.; Papathoma-Köhle, M. 2019. Recent advances in vulnerability assessment for the built environment exposed to torrential hazards: Challenges and the way forward. Journal of Hydrology, 575: 587-595. https://doi.org/10.1016/j.jhydrol.2019.05.067

Dottori, F., Figueiredo, R., Martina, M. L. V., Molinari, D., and Scorzini, A. R.: INSYDE: A synthetic, probabilistic flood damage model based on explicit cost analysis, Nat. Hazards Earth Syst. Sci., 16, 2577–2591, https://doi.org/10.5194/nhess-16-2577-2016, 2016.

Line 216: "by building footprint"? what happens with multi-storey buildings?

*Reply: Here we considered the building footprint to be in line with the INSYDE model that supplies damage by referring only to flooded floors (including basement, if present) (Dottori et al., 2016; Molinari and Scorzini, 2017)*

Dottori, F., Figueiredo, R., Martina, M. L. V., Molinari, D., and Scorzini, A. R.: INSYDE: A synthetic, probabilistic flood damage model based on explicit cost analysis, Nat. Hazards Earth Syst. Sci., 16, 2577–2591, https://doi.org/10.5194/nhess-16-2577-2016, 2016.

Molinari, D. and Scorzini A. R.: On the Influence of Input Data Quality to Flood Damage Estimation: The Performance of the INSYDE Model, Water, 9, 688, https://doi.org/10.3390/w9090688, 2017.

Lines 245-262: I find this part very difficult to follow. Again, here, a methodological workflow would be helpful.

*Reply: Authors rearranged the methodological workflow merging Figure 2 and Figure 3 to make the methodology framework clearer.*

Line 253: Who decides these classes (High, Low, Medium etc.) and under what criteria?Who is the receptor of these results (and the potential end-user of the study?) What are their needs and how are these covered by the results?

*Reply: Vulnerability classes are defined using the Jenks Natural Breaks algorithm. With natural breaks classification (Jenks) Natural Breaks Jenks, classes are based on natural groupings inherent in the data. Class breaks are created in a way that best groups similar values together and maximizes the differences between classes. The features are divided into classes whose boundaries are set where there are relatively big differences in the data values. Receptors of the results and potential end users have been better clarified throughout the manuscript, in the Introduction, the Discussion and Conclusions sections. Specifically, in subparagraph 5.3 we stressed the users'needs covered by the study and relative outcomes.*

Lines 271-273: How do the authors interpret these results?

*Reply: As It can be seen by observing the depth–damage functions for the residential sector the shape of the damage for the 2014 flood is steeper to 0.2 m water depth than other functions. Nearly maximum damage occurs when water depths exceed 0.5 m. This might be interpreted as observing the 'Very High' and 'High' vulnerable building distribution against water depth classes. Mostly vulnerable buildings fell within 0.26-0.50 m class (see the frequency distribution for the 2014 event and the three flood scenarios in the histogram below).*

[Figure]

Lines 280-283: How do the authors interpret these results?

*Reply: The shape of the damage for the 'Continuous urban fabric (S.L. : > 80 %) is similar for the 500 and 100 years return period scenarios. Whereas, for the 10 years return period scenario and for the 2014 flood the functions are much steeper in the first meter and in the first 0.2 m of water depth*

*respectively. As It can be seen by looking at Figure 9, for 'Continuous urban fabric (S.L.:> 80%)' there are more areas affected by low flood depth (less than 0.2-0.5 m) for the 2014 food and 10 years return period than the other two scenarios. It is worth to be noticed that the damage factor on built up areas is different for diverse types of land use (Vanneuville et al., 2006; Gabriels et al., 2022), but tends to be steeper within the first meter. The applied INSYDE model has shown similar conclusions (Scorzini et al., 2022). Moreover, countries around the world have proved to have similar depth–damage curves for residential buildings (Huizinga et al., 2017), again, with steeper curves within the first meter (Jongman et al., 2012).*

Vanneuville W, Maddens R, Collard C, Bogaert P, de Maeyer P, Antrop M (2006) Impact op mens en economie t.g.v. overstromingen bekeken in het licht van wijzigende hydraulische condities, omgevingsfactoren en klimatologische omstandigheden, vol MIRA/2006/02. Universiteit Gent, Vakgroep Geografie, Gent, Belgium, pp 3–121

Scorzini, A. R., Dewals, B., Rodriguez Castro, D., Archambeau, P., and Molinari, D.: INSYDE-BE: adaptation of the INSYDE model to the Walloon region (Belgium), Nat. Hazards Earth Syst. Sci., 22, 1743–1761, https://doi.org/10.5194/nhess-22-1743-2022, 2022.

Gabriels K., Willems P., and Van Orshoven J.: A comparative flood damage and risk impact assessment of land use changes, Natural Hazards and Earth System Sciences,Feb 11,22(2): 395-410, https://doi.org/10.5194/nhess-2021-51, 2022

Jongman, B., Kreibich, H., Apel, H., Barredo, J. I., Bates, P. D., Feyen, L., Gericke, A., Neal, J., Aerts, J. C. J. H., and Ward, P. J.: Comparative flood damage model assessment: towards a European approach, Nat. Hazards Earth Syst. Sci., 12, 3733–3752, https://doi.org/10.5194/nhess-12-3733-2012, 2012.

Huizinga, J., De Moel, H. and Szewczyk, W.: Global flood depth-damage functions: Methodology and the database with guidelines, EUR 28552 EN, Publications Office of the European Union, Luxembourg, ISBN 978-92-79-67781-6, doi:10.2760/16510, 2017.

Line 284: "buildings falling within 'Other roads and associated land'" How do buildings belong to a category named "roads".

***Reply:*** *Following the Urban Atlas description of LU/LC thematic classes, 'other roads and associated land' class includes roads, crossings, intersections and parking areas, including roundabouts and sealed areas with "road surface". The minimum mapping width (MinMW) between 2 objects for distinct mapping is 6 m to maintain continuity of linear structures as roads (see* https://land.copernicus.eu/user-corner/technical library/urban_atlas_2012_2018_mapping_guide*). The road network in urban areas owing to the high proportion of impermeable surfaces that prevent the infiltration of water into the soil and flood events occurrence might cause water overflow that can result in drains exceeding their capacity and flood velocity rise. Therefore, buildings fallen in this class can experience higher damages given their proximity to the road network. Authors considered maintaining this spatially detailed classification of land use distribution valuable.*

Line 290-291: "making a distinction between elements with and without basement" why is this distinction necessary?

***Reply:*** *In the Italian context, especially in urbanized areas, many buildings have an underground cellar or basement usually used as a residence or underground garage (Arrighi et al., 2020). In highly flood-prone areas basement flooding can actively infer on buildings damage (Molinari et al., 2020). If wastewater enters and flows toward the basement and if drainage systems are inadequate and a good preventive maintenance plan is missed, water can slowly seep in through cracks and holes in the foundation causing potential long-term structural damages. Furthermore, in the Italian context*

*several planners and policy makers at local level recognized the presence of the basement as a critical issue especially in urban areas pointing out the paucity of this data when dealing with buildings vulnerability assessment.*

Arrighi, C., Mazzanti, B., Pistone, F., and Castelli, F.: Empirical flash flood vulnerability functions for residential buildings, SN Appl. Sci., 2, 1–12, https://doi.org/10.1007/s42452-020-2696-1, 2020.

Molinari, D., Rita Scorzini, A., Arrighi, C., Carisi, F., Castelli, F., Domeneghetti, A., Gallazzi, A., Galliani, M., Grelot, F., Kellermann, P., Kreibich, H., Mohor, G. S., Mosimann, M., Natho, S., Richert, C., Schroeter, K., Thieken, A. H., Paul Zischg, A., and Ballio, F.: Are flood damage models converging to "reality"? Lessons learnt from a blind test, 20, 2997–3017, https://doi.org/10.5194/nhess-20-2997-2020, 2020.

Line 371: In which way was the understanding of vulnerability improved?

*Reply: The authors characterized flood hazard intensity on the basis of variability in water depth during a recent event and spatial exposure also as a function of building surrounding and buildings intrinsic characteristics as a determinant factor of the element at risk susceptibility and response capacity. In this sense the use of a geographic scale sufficient to depict spatial differences in vulnerability from land use to building level allows to identify structural vulnerability hotspots and to inform depth-damage curve for calculating potential losses.*

Line 377-378: The authors should elaborate more here. Reference to existing curves is also missing. This is a good source of the recent advances in the field:

Fuchs, S.; Keiler, M.; Ortlepp, R.; Schinke, R.; Papathoma-Köhle, M. 2019. Recent advances in vulnerability assessment for the built environment exposed to torrential hazards: Challenges and the way forward. *Journal of Hydrology,* 575: 587-595. https://doi.org/10.1016/j.jhydrol.2019.05.067

*Reply: Authors elaborated more here as suggested by the Referee as follows:*

*'In Italy the current national and regional databases are often of insufficient quality to support a robust analysis. One of the three main elements to be correlated among hazard, vulnerability, and losses is often missing or too hazy to make an appropriate comparison with scientific findings (Molinari et al., 2014). Here effects of hazard interactions on the buildings susceptibility and exposure is hazard-intensity specific based on 2014 flood event severity, however more empirical data should be included to provide more powerful analytical tools (e.g., vulnerability curves). Several studies have shown that estimations based on depth-damage functions may be very uncertain since water depth and building use only represent a fraction of the whole data variance (Merz et al. 2004). Therefore, the lack of loss data or scattered information on the damage suffered by buildings do not allow a proper comparative assessment between past damage flood events data to those findings obtained by modelled depth-damage functions. As well better information on past events and damage occurrence and amount would provide more information regarding the building response capacity implying detailed event documentation (Fuchs et al., 2019)'*

Line 377-383: The limitations part of the paper is very short. All the assumptions and limitations have to be acknowledged.

*Reply: The authors provided a new subparagraph 5.3 Advantages, limitations and future development where main study's limitations have been described in greater detail and used as starting point for future research development proposals as follows:*

*'Nevertheless, some limitations can be noted. Considerable amount of data at local scale is needed, however it is not always available to the local authorities or cannot be accessible or easily collected.*

*In Italy the current national and regional databases are often of insufficient quality to support a robust analysis. One of the three main elements to be correlated among hazard, vulnerability, and losses is often missing or too hazy to make an appropriate comparison with scientific findings (Molinari et al., 2014). Here effects of hazard interactions on the buildings susceptibility and exposure is hazard-intensity specific based on 2014 flood event severity, however to provide more powerful analytical tools more empirical data should be included. Several studies have shown that estimations based on stage-damage functions may be very uncertain since water depth and building use only represent a fraction of the whole data variance (Merz et al. 2004). Therefore, the lack of loss data or scattered information on the damage suffered by buildings do not allow a proper comparative assessment between past damage flood events data to those findings obtained by modelled depth-damage functions. As well, better information on past events and damage occurrence and amount would provide more information regarding the building response capacity and would improve heuristic approach and susceptibility assessment. Thus, the scoring procedure in this study has been done on basis of literature having many associated uncertainties.'*

Conclusions: there is a focus on vulnerability here whereas the paper seemed to have as an aim the calculation of loss. Furthermore, there is no discussion about possible uses and end-users of the presented research.

**Reply:** *As stated before the authors provided a new subparagraph 5.3 Advantages, limitations and future development where potential uses and possible end-users have been described.*

*'The study presents several advantages to support decision makers, as planners and policy makers, in improving their investment strategies for the mitigation and the reduction of flood damages, but also to improve the currently low flood insurance coverage exploring new insurance model trials to sustain insurance system for residential properties and (Gizzi et al., 2016). The SPRC model is a valuable tool to support decision makers in better understanding vulnerability drivers, complex processes and their interrelations acting as a guide for interventions, priority measures and resources allocation either in the pre or post- event phases. Examples of pre-flood applications of the outputs may include the development of flood hazard mitigation strategies that outline policies and programs for reducing flood losses, including nature-based solutions or the use of the obtained AAL as an input of a cost-benefits analysis of prevention and mitigation measures. Thereby examples of post-event applications of the outputs may include the application of land use planning principles and practices, the allocation of resources for flood-resilient buildings interventions.'*

*Comments on the Figures:*

Figure 5: Attention: in the map legend it is NISR instead of NIRS.

**Reply:** *Changed.*

Figure 7: Why this choice of colours? For example, high buildings should be less vulnerable and experience a lower degree of loss (red colour) and the low buildings that should be more vulnerable are green.

**Reply:** *As pointed out by the Referee the figure legend was wrong. Authors corrected Figures 7.*

Figure 7d: what is the difference between apartment, residential and attached/semidetached? Are they not all residential?

**Reply:** *Here we refers to OpenStreetMap (OSM) definitions of residential buildings:*
*Apartments: It is a housing unit which is part of a building (block of apartment) with several floor levels. May also have retail outlets on the ground floor;*

*Detached/semi-detached houses: A dwelling unit inhabited by a single household or It is a duplex dwelling house which shares a common side with the next house;*
*Residential buildings: buildings with more than one floor used primarily for residential purposes that don't fall into previous classes (including attached house, a type of dwelling which is in a block with several housing units, one next to the other).*

Figure 7f: Building values are uniform? Is it because they are the market values? So, they have nothing to do with the physical vulnerability in my opinion.

*Reply: Here market value is used to define the monetary exposure of the building and it has been analysed at single-building scale according to the building type and building status (Arrighi et al., 2020). In figure 7f the map shows the market value reference zone according to the National Real Estate Observatory (OMI) classification based on territory sub-division (central and semi-central urban areas, suburban, peri-urban and peripheral areas) having higher market values quotation (increasing from red to green) (*[https ://wwwt.agenz iaent rate.gov.it/geopo i_omi/index.php](https ://wwwt.agenz iaent rate.gov.it/geopo i_omi/index.php)*) and the percentages of residential buildings falling within OMI zones (pie chart), hence it is not the economic unit value in € m$^{-2}$ that is not homogeneous. The average, minimum and maximum market unit value for each residential building in € m$^{-2}$is reported in Table 4 and related to the obtained vulnerability classes. For the loss estimation damage are modelled by means INSYDE damage functions that refers to the reparation of the damaged elements or to their removal and replacement costs derived from reference price lists (Dottori et al., 2016).*
*Figure 7f caption has been changed in :*
*'(f) OMI zones map according to the National Real Estate Observatory (OMI) classification based on territory sub-division (central and semi-central urban areas, suburban, peri-urban and peripheral areas) having higher market values quotation (increasing from red to green).'*

Arrighi, C., Mazzanti, B., Pistone, F., and Castelli, F.: Empirical flash flood vulnerability functions for residential buildings, SN Appl. Sci., 2, 1–12, https://doi.org/10.1007/s42452-020-2696-1, 2020.

Dottori, F., Figueiredo, R., Martina, M. L. V., Molinari, D., and Scorzini, A. R.: INSYDE: A synthetic, probabilistic flood damage model based on explicit cost analysis, Nat. Hazards Earth Syst. Sci., 16, 2577–2591, https://doi.org/10.5194/nhess-16-2577-2016, 2016

Figure 8: Similar comment about the choice of colours for buildings with and without a basement.

*Reply: The colour here is firstly assigned on the basis of vulnerability class (i.e., orange for the 'High' vulnerability and red for the 'Very High' vulnerability classes respectively), secondly darker hue of orange and red has been used to identify the presence of basement respectively, emphasising buildings higher susceptibility.*

Figure 8: what are "other buildings" and "other vulnerability class"?
*Reply: 'Other buildings' class includes no residential buildings (e.g., commercial, public facilities, industrial). In figure 8b and 8d vulnerability hotspots are identified and most vulnerable residential buildings with and without basement map are shown, thereby "other vulnerability class" refers to 'Moderate', 'Low' and 'Very Low' vulnerability classes.*

Figure 9: The difference between the three maps is minimal. Is it because it is only a smaller part of a larger inundation map? How do the authors explain this similarity?

*Reply: The area considered here is part of a larger inundation map. That is why slight changes occurred in this area compared to the entire modelled extent. However, comparing the three maps one can see where major and mostly localised changes occurred. Flood depths for different return*

*periods have generally small differences, due to the relatively flat area where the floodwaters can spread. Localised spots with flood depths larger than 4 metres occur in depressions being filled by the floodwaters (e.g., metro stations, road underpasses). Because of the need to visualize these large flood depths in the maps of Figure 9, smaller differences due to increasing return periods are less visible in the figure.*

Figures 10, 11: damage curves with absolute damage make no sense to me. What kind of information do they deliver that can be useful? Give more information about the making of the curves. Which regression model did the authors use? (Weibull distribution, logistic?).

*Reply: In the INSYDE model the selection of synthetic functions assumed that all properties are equal (Martínez-Gomariz et al. 2020) and adopt an expert-based approach about damage mechanisms (Dottori et al. 2016; Malgwi et al. 2021) by selecting the benefits of a multivariable model, stated by Amadio et al. (2019), Schröter et al. (2014) and Wagenaar et al. (2018), which require a comprehensive set of data in order to correctly identify complex relationships among variables. INSYDE model can estimate relative damage (i.e. percentage estimation of losses with respect to the total value of the building) and absolute damage, the latter considers the unit prices of cost of damage. Here we decide to express the damage in absolute terms to give the monetary measure using the cost per unit of measure (e.g., square meter).*

Amadio M, Scorzini AR, Carisi F, Essenfelder AH, Domeneghetti A, Mysiak J and Castellarin A.: Testing empirical and synthetic flood damage models: the case of Italy, Natural Hazards and Earth System Sciences, Mar 29, 19-3, 661-78, https://doi.org/10.5194/nhess-19-661-2019, 2019

Dottori, F., Figueiredo, R., Martina, M. L. V., Molinari, D., and Scorzini, A. R.: INSYDE: A synthetic, probabilistic flood damage model based on explicit cost analysis, Nat. Hazards Earth Syst. Sci., 16, 2577–2591, https://doi.org/10.5194/nhess-16-2577-2016, 2016.

Malgwi MB, Schlögl M and Keiler M.: Expert-based versus data-driven flood damage models: A comparative evaluation for data-scarce regions, International journal of disaster risk reduction, Apr 15, 57, 102148, https://doi.org/10.1016/j.ijdrr.2021, 2021.

Martínez-Gomariz, E., Forero-Ortiz, E., Guerrero-Hidalga, M., Castán, S. and Gómez, M.: Flood Depth–Damage Curves for Spanish Urban Areas. Sustainability, 12-7, p.2666, https://doi.org/10.3390/su12072666, 2020.

Schröter K, Kreibich H, Vogel K, Riggelsen C, Scherbaum F and Merz B.: How useful are complex flood damage models?, Water Resources Research, 50-4, 3378-95, https://doi.org/10.1002/2013WR014396, 2014

Wagenaar D, Lüdtke S, Schröter K, Bouwer LM and Kreibich H.: Regional and temporal transferability of multivariable flood damage models, Water Resources Research, 54-5, 3688-703, https://doi.org/10.1029/2017WR022233, 2018.

How do the authors interpret the fact that after a point the curves are steady? Why do they not extend beyond a specific water depth? What is the role of the number of floors?And most importantly: what are the potential uses and end-users of these curves?

*Reply: In Figure 10 b, differences between the curves are related to the extension and intensity of the return periods and the studied 2014 event. Curves flattens off rather abruptly after 1 m for all return periods, and 0.5 m for the 2014 flood event. This pattern has been found in other works (Freni et al. 2010; Pristika et al. 2014; Albano et al. 2014; Romali and Yusop 2021). The main reason is related to the asset's replacement cost, which does not rise so much once a certain water level is*

*achieved. The number of floors is one of the indicators that work as an input for the INSYDE model, however it does not significantly affect the present results. One floor in Italy is equivalent to 2.40-3 m (Italian Ministerial Order 1975[1]), which actually are close to the highest values, where the curve is already steady. Decision makers, urban planners and users that deal with flood insurance might be potential end users of these curves. Good strategies and actions against disasters should be a result of better understanding of disaster risk (Cardona et al., 2018), thereby understanding where the most vulnerable areas are located considering the level of damage in different parts of the city could have several potential uses such as:*

1. *to produce a more accurate early warning system (EWS)*
2. *to accurately define evacuation sites*
3. *to prioritize areas for better risk mitigation planning*
4. *to improve the currently low flood insurance coverage*

Albano R, Sole A, Sdao F, Giosa L, Cantisani A, and Pascale S.: A systemic approach to evaluate the flood vulnerability for an urban study case in Southern Italy, Journal of Water Resource and Protection, Mar 20, http://dx.doi.org/10.4236/jwarp.2014.64037, 2014.

Cardona, O.D., et al.: Determinants of risk: Exposure and vulnerability. In: Field, CB, Barros V, Stocker TF, Qin D, Dokken DJ, Ebi KL, T Welle and J Birkmann 1550003-32 Mastrandrea MD, Mach KJ, Plattner G-K, Allen SK, Tignor M and Midgley PM (eds.) Managing the Risks of Extreme Events and Disasters to Advance Climate Change Adaptation. A Special Report of Working Groups I and II of the Intergovernmental Panel on Climate Change (IPCC), Cambridge, UK, and New York, NY, USA: Cambridge University Press, 65–108, https://doi.org/10.1017/CBO9781139177245.005, 2012.

Freni G, La Loggia G and Notaro V.: Uncertainty in urban flood damage assessment due to urban drainage modelling and depth-damage curve estimation, Water Science and Technology, 61-12, 2979-93, http://dx.doi.org/10.2166/wst.2010.177, 2010

Pistrika A, Tsakiris G. and Nalbantis I.: Flood depth-damage functions for built environment, Environmental Processes,1-4,553-7, https://doi.org/10.1007/s40710-014-0038-2, 2014.

Romali NS and Yusop Z.: Flood damage and risk assessment for urban area in Malaysia, Hydrology Research, Feb 1;52-1:142-59, http://dx.doi.org/10.2166/nh.2020.121, 2021.
* * *
[1] https://www.certifico.com/costruzioni/379-documenti-costruzioni/documenti-riservati-costruzioni/8149-altezza-locali-abitativi-norme-e-valori

---

## Author Comment (AC2)

**Comment on egusphere-2022-225**

**Answer to Anonymous Referee #3**

This paper presents an approach to calculate flood losses to residential buildings using a Source-Pathway-Receptor-Consequence model. The proposed approach is applied to an area of Milan, Italy which is frequently impacted by flood events. The authors apply the approach to a historical flood event that occurred in 2014 and three flood scenarios. The vulnerability of residential buildings is mapped, and damage curves are derived. The proposed approach has merit and the results are interesting, however, the paper is at times hard to follow and would benefit from some changes before publication. I have outlined my suggestions below.

*We thank the reviewer for his careful reading of the manuscript and his constructive remarks. We have taken the comments on board to improve and clarify the manuscript. The vast majority of those have resulted in an addition to the text or to a change. Please find below a detailed point-by-point response to all comments (Referee's comments in black, our replies in red).*
*We thank the reviewer for his careful reading of the manuscript and his constructive remarks. We have taken the comments on board to improve and clarify the manuscript. The vast majority of those have resulted in an addition to the text or to a change. Please find below a detailed point-by-point response to all comments (Referee's comments in black, our replies in red).*
*Major general changes:*
*• Modify the abstract to reflect clearer the scope of the paper;*
*• Added a clearer 'Background' and 'Aims' sections;*
*• Provided more detail on the Methodology throughout the manuscript;*
*• Ensuring a consistent results interpretation in the Discussion;*
*• Enhanced the figures.*

**Specific Comments**

Figure 1: what does the blue outline represent in the right panel in Figure 1? Is it the area of inundation or the study boundary? It would be useful to add a label describing this area to the map legend.

**Reply:** *The blue outline in the right panel of Figure 1 represents the 2014 flooded area that also corresponds to the study boundary. Following the Referee advice the authors added a label to the map legend.*

Figure 2: this figure needs to be improved. Only the source and the pathway symbols are easily understood. It is not clear what the receptor or consequence arrows are referring too. At the very least, the figure description needs to be improved so that the image can be understood on its own. I would suggest also improving the figure to make it more understandable.

**Reply:** *Figure 2 has been merged with Figure 3 to make the workflow of the SPRC model clearer and for consistency concerning to the study applied methodology.*

Line 134: What digital terrain model are you using?

*Reply: We have added that "FwDET identifies the floodwater elevation for each cell within a flooded domain based on its nearest flood-boundary grid cell here derived from the Digital Terrain Model (DTM) of the Lombardy Region with a resolution of 5 by 5 meters (Bocci et al., 2015)."*

Line 139: "This model analyses the DTM with hydrology…". What does this mean? Please expand on the approach.

*Reply: The sentence has been rephrased for clarity and a brief model description has been included as follows:*

*'This model is based on the screening of a digital terrain model for landscape depressions and their maximum extent when filled up at the capacity before spilling over during a flood while ignoring local infiltration rates and time, thereby allowing the model to select buildings inside or adjacent to these low-lying areas'*

Line 143: resampled using what approach? From what resolution?

*Reply: Resampling approach and layers resolution have been included as follows:*

*'Therefore, the Copernicus High Resolution Layer Imperviousness Degree 2018 with a resolution of 20 m resampled by nearest-neighbours at 5 m is used to identify most exposed residential buildings'*

Equation 1: Is the denominator supposed to be Building Footprint Area?

*Reply: Yes, it is. We change the denominator from 'Building Footprint' to 'Building Footprint Area' for clarity.*

Line 179: What is the reasoning behind assigning a building with a basement a more favourable weight than a building without a basement?

*Reply: The assignment of weights refers to the building's response capacity against flood hazard, meaning the capability or incapability of an object to resist the flood impact. Hence, lower values correspond to lower response capacity (high vulnerable buildings), whereas higher values correspond to higher response capacity (low vulnerable buildings).*

Section 3.2.1: How sensitive is the final score (and thus your outcomes) to the assumptions you make about the weights? For example, the choice of how many weights you assign to each factor has an impact on results. Are these weights taken directly from the literature? It would be good if the authors could elaborate a bit further on this.

*Reply: The assignment of weights for the heuristic approach refers to the building's response capacity against flood hazard, meaning the capability or incapability of an object to resist the flood impact. Hence, lower values correspond to lower response capacity (high vulnerable buildings), whereas higher values correspond to higher response capacity (low vulnerable buildings)(Table 2). The weights assignment have been given by authors based on an intensive literature review and on data availability. For indicators like Construction Material, Period of Construction, Building Status ranges like 1-9 or 1-4 were chosen, following the study of Corradi et al. (2015). In virtue of the possible methods (statistical assessments such as Principal Component Analysis, factor analysis, or participatory approaches like Analytic Hierarchy Process or Budget Allocation Process) an approach using equal weights by Taramelli et al. (2015) was selected, which represents the most adopted*

*approach in literature (Beccari 2016; Papathoma-Köhle 2019). These indicators are identified as significant and suitable vulnerability indicators in urban areas representative of three components, i.e., flood hazard intensity, effect of the surrounding environment and building characteristics. A final total score for the heuristic approach is calculated by summing up all the weights assigned to each indicator composing a residential building, and classifying them in 5 categories, from 'Very High', to 'Very Low' vulnerability (Rincon et al. 2018, Gatti, 2020) using the Jenks Natural Breaks algorithm. With natural breaks classification (Jenks) Natural Breaks Jenks, classes are based on natural groupings inherent in the data. Class breaks are created in a way that best groups similar values together and maximizes the differences between classes. In addition, a different consideration was made to buildings with and without a basement. Here the aim was to set up the main vulnerability indicators based on the SPRC chain based on hazard intensity, building surrounding and buildings intrinsic characteristics. Overall, because of the method selected, the model output is quite sensitive to assumed levels of parameters. A robust sensitivity analysis would be a further step.*
*Furthermore additional and detailed information have been also added to the manuscript as follow:*

*'Age and maintenance are also indications for the current state of the building. Moreover, an estimation of elements-at-risk costs is fundamental to express losses in economic terms. The heuristic approach is based on a simple equal weights assignment procedure (Taramelli et al., 2015). The weights assignment have been given by authors based on an intensive literature review and on data availability. The indicators are identified as significant and suitable vulnerability indicators in urban areas representative of three components, i.e., flood hazard intensity, effect of the surrounding environment and building characteristics.'*

*'For buildings falling in most vulnerable classes (i.e., class 1 and 2, 'Very High' and 'High') a distinction was made between elements with and without basement (Arrighi et al., 2020; Molinari et al., 2020) assigning a weight of 0 to the building with basement and 1 to the building without basement (McBean et al., 1988; Crigg and Helweg, 1975). As previously mentioned, the assignment of weights refers to the building's response capacity against flood hazard, meaning the capability or incapability of an object to resist the flood impact. Hence, lower values correspond to lower response capacity (high vulnerable buildings), whereas higher values correspond to higher response capacity (low vulnerable buildings) (Taramelli et al., 2015).'*

Beccari, B.: A comparative analysis of disaster risk, vulnerability and resilience composite indicators." PLoS currents, 8, https://doi.org/10.1371/currents.dis.453df025e34b682e9737f95070f9b970, 2016.

Corradi, J., Salvucci, G., and Vitale, V.: Analisi della vulnerabilità sismica dell'edificato italiano : tra demografia e " domografia " una proposta metodologica innovativa, 1–22, 2015.

Gatti, I.: Disaster risk assessment for urban areas : A GIS flood risk analysis for Luján City (Argentina), M.Sc. thesis, The University of Tokyo, Japan, 75pp, 2020.

Papathoma-Köhle M, Cristofari G, Wenk M and Fuchs S: The importance of indicator weights for vulnerability indices and implications for decision making in disaster management, International journal of disaster risk reduction, 36,101103, https://doi.org/10.1016/j.ijdrr.2019.101103, 2019.

Rincón, D., Khan, U.T. and Armenakis, C.: Flood risk mapping using GIS and multi-criteria analysis: A greater Toronto area case study, Geosciences, 8, 275, https://doi.org/10.3390/geosciences8080275, 2018.

Taramelli, A., Valentini, E., and Sterlacchini, S.: A GIS-based approach for hurricane hazard and vulnerability assessment in the Cayman Islands, Ocean Coast. Manag., 108, 116–130, https://doi.org/10.1016/j.ocecoaman.2014.07.021, 2015.

Figure 6: I can't make out what categories half the box plots belong too because they are so small. Also, the text is too small to be read on the smaller maps. Please adjust the figure.

*Reply: The figure has been improved to make it clearer.*

**Technical Errors**

Line 25: consider replacing "it" with "one"

*Reply: Done.*

Line 40: "location" should be "location's"

*Reply: Changed.*

Lines 64-66: Sentence beginning "Vamvatsikos" and ending "various scales". I would consider rephrasing this for clarity

*Reply: The sentence has been rephrased for clarity as suggested by the Referee as follows:*

*'Some authors stressed the need of the use of empirical data from past event to provide powerful analytical tools (Vamvatsikos et al., 2010; Apel et al., 2008).'*

Lines 135-134: Sentence beginning "Specifically," and ending "near them". I would consider rephrasing this for clarity

*Reply: Authors have rephrased the sentence as suggested by the Referee as follows:*

*'We investigated endanger urban residential areas located within or near landscape sinks (SI) that are potentially filled in conditions of flooding and inefficient drainage system (Dietrich and Perron, 2006; Dodov and Foufoula-Georgiou, 2006; Nardi et al., 2006; Taramelli and Reichenbach, 2008; Thrysøe et al., 2021).'*

Equation 2: Period of Construction is shortened to "PC" in the equation but to "PT" in the preceding text. Change one to maintain consistency.

*Reply: Authors changed the abbreviation from "PC" to "PT" in the preceding text.*

Line 212: "Fig.1C " in the text is labelled "Fig C1" in the Appendix. Please change for consistency

*Reply: Authors changed in Fig. C1.*

Figure 10: It seems the labels for figure 10b and Figure 10c are the wrong way around.

*Reply: Caption of Figure 10 has been corrected as follows:*

*'Figure 10: (a) Absolute Damage for the residential sector (n° of buildings) considering 500, 100 and 10 years of return period and the 2014 flood event; (b) Site-specific depth–damage curve for the residential sector considering 500, 100 and 10 years of return period and the 2014 flood event; (c) Exceedance probability of Absolute Damage for the residential sector.'*